# A COOPERATIVE-GAME-THEORETICAL MODEL FOR AD HOC TEAMWORK

## ABSTRACT

Ad hoc teamwork (AHT) is a cutting-edge problem in the multi-agent systems community, in which our task is to control an agent ('learner') which is required to cooperate with new teammates without prior coordination. Prior works formulated AHT as a stochastic Bayesian game (SBG), standing by the view of non-cooperative game theory. Follow-up work extended SBG to open team settings and proposed an empirical implementation framework based on GNNs called Graph-based Policy Learning (GPL) to tackle variant team sizes. Although the performance of GPL is convincing, its global Q-value representation is difficult to interpret and, therefore, impedes the potential application to real-world problems. In this work, we introduce a game model called coalitional affinity game (CAG) in cooperative game theory and establish a novel theoretical model named open stochastic Bayesian CAG to describe the process of AHT with open team settings. Based on the theoretical model, we derive the new solution concept that guides the representation of the global Q-value with theoretical guarantees for this setting. We further design a practical algorithm which can easily implement the theoretical results. In experiments, we demonstrate the performance improvement the proposed algorithm over GPL and verify the effectiveness and reasonableness of our theoretical model. The demo of the experiments is available at `https://sites.google.com/view/cagpl/`.

## 1 INTRODUCTION

Ad hoc teamwork (AHT) is a cutting-edge problem in the multi-agent system community (Stone et al., 2010) to investigate how an agent could be trained to cooperate on the fly with other teammates without prior coordination methods (e.g., via centralized training and shared tasks) (Mirsky et al., 2022). Such scenarios often appear in real-world problems. In search and rescue, a robot must collaborate with other unknown robots (e.g., manufactured by various companies without a common coordination protocol (Barrett & Stone, 2015)) or humans to rescue survivors. Similar situations may occur in AI that helps trading markets (Albrecht & Ramamoorthy, 2013), as well as in the human-machine / machine-machine collaboration emerging from the prevailing embodied AI (Smith & Gasser, 2005; Duan et al., 2022) and applications of large language models (LLMs) (Brown et al., 2020; Zhao et al., 2023).

Unlike traditional multi-agent reinforcement learning (MARL), AHT only allows controlling one agent, the learner, to adapt to its ad hoc teammates to complete a task with a common goal. Prior work formulated AHT as a stochastic Bayesian game (SBG) (Albrecht & Ramamoorthy, 2013) from the perspective of non-cooperative game theory, considering the uncertainty of teammates by introducing the Bayesian game (Harsanyi, 1967) and long-term collaboration by introducing a stochastic game (a.k.a. Markov game) (Shapley, 1953). The uncertainty of agents in a Bayesian game is characterized by types (i.e., each agent is assumed to belong to an instance of a set of types), which is usually called the type-based method in the context of AHT (Mirsky et al., 2022). Rahman et al. (2021) extended the SBG to the open team setting (i.e., team size would vary over time) named open stochastic Bayesian game (OSBG). The open team setting is usually called open ad hoc teamwork (OAHT). To implement OAHT, a framework called Graph-based Policy Learning (GPL) was proposed with a specific representation of the global Q-value and graph neural networks (GNNs) to address the open team setting. *Albeit the effectiveness in experiments (almost state-of-the-art), GPL lacks a theoretical*

*interpretation to show its reliability. This impedes potential deployment to real-world problems, which necessarily requires reliability and interpretability of algorithms (Wang et al., 2021).*

To address the above issue, we introduce cooperative game theory to establish a novel theoretical model to describe OAHT and design an algorithm with theoretical guarantees. The contributions of this work can be summarized as follows. We first introduce a game model from cooperative game theory called coalitional affinity games (CAG) (Brânzei & Larson, 2009) and incorporate it into OSBG with detailed specifications that are lacking in the previous work (Rahman et al., 2021) to form a new game model named open stochastic Bayesian coalitional affinity games (OSB-CAG). Different from the non-cooperative game theory, we treat the achievement of a common goal as the problem how an open ad hoc team can be formed to collaborate in a stable manner (i.e., no agents would have incentives to leave) under the learner's influence on teammates through decision making. To formalize the learner's influence on ad hoc teammates, we define the relationship between them as a star affinity graph, where the learner is the internal node and teammates are leaves, in light of the definition of AHT that teammates have no prior coordination and therefore no explicit relationship. The number of leaves at each timestep varies depending on the size of the open ad hoc team. As per the OSB-CAG theory, each agent's preference Q-value (depicting its incentive to leave) is naturally defined as the sum of pairwise and weighted individual utility terms. The solution concept named ad hoc Bayesian strict core (AH-BSC) describing the stability of ad hoc teams is proved to be reached when the sum of agents' preference Q-values as the global Q-value is maximized. This deepens understanding of the significance of value representation utilized in GPL for encouraging ad hoc teamwork. Additionally, the OSB-CAG theory suggests an inductive bias of the components constituting the global Q-value (i.e. the pairwise and individual utility terms) to facilitate learning.

To solve AH-BSC, we transform it to the open ad hoc Bellman optimality equation and propose the corresponding Bellman operator named open ad hoc Bellman operator, which is further transformed to Q-learning training loss. Thanks to the OSB-CAG theory, we propose an empirical method implemented based on GPL named as coalitional affinity GPL (CA-GPL). In experiments, we mainly compare CA-GPL with GPL and some variants of CA-GPL on Level-based foraging (LBF) and Wolfpack with open team settings (in both previously unseen compositions and team sizes) (Rahman et al., 2021; 2022), to validate the reasonableness and effectiveness of our theoretical model. Due to the page limits, we move related works to Appendix A.

## 2 BACKGROUND

In this paper, let $\Delta(\Omega)$ indicate the set of probability distributions over the random variable on a sample space $\Omega$ and let $\mathbb{P}$ denote the power set. To simplify the notation, let $i$ denote the learner, and $-i$ indicates the joint set of its teammates at timestep $t$. $P(\mathcal{X})$ indicates the generic probability distribution over a random variable $\mathcal{X}$. $|\mathcal{X}|$ indicates the cardinality of a set $\mathcal{X}$ in this work.

### 2.1 COALITIONAL AFFINITY GAMES

As a subclass of non-transferable utility (NTU) games, hedonic game (Chalkiadakis et al., 2022) is defined as a tuple $\langle \mathcal{N}, \succeq \rangle$ in this paper, where $\mathcal{N} = [n]$ is a finite set of agents; and $\succeq = (\succeq_1, ..., \succeq_n)$ is a list of agents' preferences over corresponding subsets of $\mathcal{N}$ called coalitions. $\mathcal{C} \succeq_j \mathcal{C}'$ means the preference of agent $j$ over coalition $\mathcal{C}$ is no less preferred than coalition $\mathcal{C}'$. For each agent $j \in \mathcal{N}$, $\succeq_j$ depicts a complete and transitive preference relation over a collection of all feasible coalitions $\mathcal{N}(j) = \{\mathcal{C} \subseteq \mathcal{N} | j \in \mathcal{C}\}$. The outcome of a hedonic game is a coalition structure $\mathcal{CS}$, i.e., a partition of $\mathcal{N}$ into disjoint coalitions. We denote $\mathcal{CS}(j)$ as the coalition including agent $j$. The ordinal preferences are represented as the cardinal form as preference values (Sliwinski & Zick, 2017) to ease learning. More specifically, each agent $j$ has a preference value function $v_j : \mathcal{N}(j) \rightarrow \mathbb{R}_{\geq 0}$. $v_j(\mathcal{C}) \geq v_j(\mathcal{C}')$ if $\mathcal{C} \succeq_j \mathcal{C}'$ that implies that agent $j$ weakly prefers $\mathcal{C}$ to $\mathcal{C}'$; $v_j(\mathcal{C}) > v_j(\mathcal{C}')$ if $\mathcal{C} \succ_j \mathcal{C}'$ that implies that agent $j$ strictly prefers $\mathcal{C}$ to $\mathcal{C}'$; and $v_j(\mathcal{C}) = 0$ if $\mathcal{C} = \varnothing$. To concisely represent the preference values, a hedonic game is equipped with an affinity graph $G = \langle \mathcal{N}, \mathcal{E} \rangle$, where each edge $(j, j') \in \mathcal{E}$ describes an affinity relation between agent $j$ and $j'$; for each edge $(j, j')$, it defines a weight $w(j, j') \in \mathbb{R}$ indicating the value that agent $j$ can receive from agent $j'$, while if $(j, j') \notin \mathcal{E}, w(j, j') = 0$. Thereby, for any $\mathcal{C} \subseteq \mathcal{N}_j, v_j(\mathcal{C}) = \sum_{(j,j') \in \mathcal{E}, j' \in \mathcal{C}} w(j, j')$ if $\mathcal{C} \neq \{j\}$, otherwise, $v_j(\mathcal{C}) = v_j(\{j\}) \geq 0$ is a constant value. We are interested in the coalitions with individual rationality for each agent $j$ such that $v_j(\mathcal{C}) \geq v_j(\{j\})$, otherwise, agents would have no incentives to

form coalitions. An affinity graph is symmetric if $w(j, j') = w(j', j)$ for all $(j, j'), (j', j) \in \mathcal{E}$. The hedonic game with an affinity graph is called a coalitional affinity game (CAG) (Brânzei & Larson, 2009). Strict core stability and inner stability are two fundamental solution concepts in the CAG, whose definition is described in Definition 1.

**Definition 1.** *We say that a blocking coalition $\mathcal{C}$ weakly blocks a coalition structure $\mathcal{CS}$ if every agent $j \in \mathcal{C}$ weakly prefers $\mathcal{C}$ to $\mathcal{CS}(j)$ and there exists at least one agent $j' \in \mathcal{C}$ who strictly prefers $\mathcal{C}$ to $\mathcal{CS}(j)$. A coalition structure admitting no weakly blocking coalition $\mathcal{C} \subseteq \mathcal{N}$, exhibits the strict core stability. On the other hand, if a coalition structure $\mathcal{CS} = \{\mathcal{C}_1, ..., \mathcal{C}_m\}$ admitting no weakly blocking coalition $\mathcal{C} \subset \mathcal{C}_k$, for some $1 \leq k \leq m$, exhibits the inner stability.*

## 2.2 General Ad Hoc Teamwork Framework

We now summarize and refine the general open ad hoc teamwork (OAHT) framework used in the past work (Rahman et al., 2021) called open stochastic Bayesian game (OSBG) which is defined as the tuple $\langle \mathcal{N}, \mathcal{S}, \mathcal{A}, \Theta, R, T, \gamma \rangle$. $\mathcal{N}$ is a set of all possible agents; $\mathcal{S}$ is a set of states; $\mathcal{A} = \times_{j \in \mathcal{N}} \mathcal{A}_j$ is a set of all possible joint actions where $\mathcal{A}_j$ is agent $j$'s action set; $\Theta$ is a set of all possible agent types. Let $\mathcal{A}_{\mathcal{N}_t} = \times_{j \in \mathcal{N}_t} \mathcal{A}_j$ be defined as the joint action set under a variable agent set $\mathcal{N}_t \subseteq \mathcal{N}$ which describes the characteristic of open team. The joint action space under the variable number of agents is defined as $\mathcal{A}_{\mathcal{N}} = \bigcup_{\mathcal{N}_t \in \mathbb{P}(\mathcal{N})} \{a | a \in \mathcal{A}_{\mathcal{N}_t}\}$, while similarly the joint agent-type space under the variable number of agents is defined as $\Theta_{\mathcal{N}} = \bigcup_{\mathcal{N}_t \in \mathbb{P}(\mathcal{N})} \{\theta | \theta \in \Theta^{|\mathcal{N}_t|}\}$. $R : \mathcal{S} \times \mathcal{A}_{\mathcal{N}} \to \mathbb{R}$ is the learner's reward. $T : \mathcal{S} \times \Theta_{\mathcal{N}} \times \mathcal{A}_{\mathcal{N}} \to \mathcal{S} \times \Theta_{\mathcal{N}}$ is transition function to describe the evolution of agents with variable types. The learner's action value function $Q^{\pi_i}(s, a_i)$ is shown as follows:

$$Q^{\pi_i}(s, a_i) = \mathbb{E}_{a_{t,-i} \sim \pi_{t,-i}} \left[ Q^\pi(s, a_i, a_{t,-i}) \right] = \mathbb{E}_{\substack{a_{t,-i} \sim \pi_{t,-i}, \\ a_{t,i} \sim \pi_i, T}} \left[ \sum_{t=0}^{\infty} \gamma^t R(s_t, a_t) \middle| s_0 = s, a_{0,i} = a_i \right], \quad (1)$$

where $\gamma \in [0, 1)$ is a discount factor; $s_t$ is a state, $a_{t,-i}$ is a joint teammates' action and $a_{t,i}$ is the ad hoc agent's action; $\pi_i$ is the learner's stationary policy and $\pi_{t,-i}$ is ad hoc teammates' joint stationary policy; $Q^\pi(s, a_i, a_{t,-i})$ is a global Q-value. The learner's policy $\pi_i^*$ is optimal, if and only if $Q^{\pi_i^*}(s, a_i) \geq Q^{\pi_i}(s, a_i)$ for all $\pi_i, s, a_i$. Teammates' joint policy is represented as $\pi_{t,-i} : \mathcal{S} \times \Theta_{\mathcal{N}} \to \Delta(\mathcal{A}_{\mathcal{N}})$. The learner is unable to observe their teammates' types and policies, which are inferred through history states and actions. The learner's decision making can be conducted by selecting actions that maximize $Q^{\pi_i}(s, a_i)$.

## 3 Open Stochastic Bayesian Coalitional Affinity Games

This section extends OSBG by considering coalitional affinity games to formalize the OAHT problem with an affinity graph and detail the representation of the global Q-value. Thereby, it is natural to model the ad hoc teamwork process as a process where the learner influences ad hoc teammates to prevent them from leaving the ad hoc team. For conciseness, we only consider the fully observable scenarios in this work, which can be further generalized to partial observation.

### 3.1 Problem Formulation

Suppose there is an environment where the learner $i$ interacts with other uncontrollable ad hoc teammates $-i$ to achieve a shared goal, and the state of the environment evolves as per a transition function. To model such a situation, we propose **O**pen **S**tochastic **B**ayesian **C**oalitional **A**ffinity **G**ames (OSB-CAG) that are described as a tuple $\langle \mathcal{N}, \mathcal{S}, \mathcal{A}, \Theta, (R_j)_{j \in \mathcal{N}}, P_T, P_I, P_A, \mathcal{E}, \gamma \rangle$. $\mathcal{N}, \mathcal{S}, \mathcal{A}$, $\Theta, \gamma, \pi_{t,-i}$ and $\pi_i$ are defined in the same way as OSBG. There exist an affinity graph $G = \langle \mathcal{N}, \mathcal{E} \rangle$, where $\mathcal{E} = \{(j, j') | j, j' \in \mathcal{N}\}$. We define three preliminary probability distributions such that $P_T : \mathbb{P}(\mathcal{N}) \times \mathcal{S} \times \mathcal{A}_{\mathcal{N}} \to \Delta(\mathbb{P}(\mathcal{N}) \times \mathcal{S})$, $P_I : \mathbb{P}(\mathcal{N}) \times \mathcal{S} \to [0, 1]$ and $P_A : \mathcal{N} \times \mathcal{S} \to \Delta(\Theta)$. These probability functions describe the dynamics of the environment: (1) At the initial timestep 0, $P_I(\mathcal{N}_0, s_0)$ generates an initial set of agents $\mathcal{N}_0$ and an initial state $s_0$; (2) $P_A(\theta_j | \{j\}, s_t)$ is a type assignment function which randomly assigns types to the generated agent set; (3) $P_T(\mathcal{N}_t, s_t | \mathcal{N}_{t-1}, s_{t-1}, a_{t-1})$ generates the agent set $\mathcal{N}_t$ and $s_t$ for the next timestep $t$; (4) Step 2 and 3 are continually repeated. To concisely represent the above process, we derive a composite transition function $T(\theta_t, s_t, \mathcal{N}_t | \theta_{t-1}, s_{t-1}, a_{t-1})$ (see Proposition 1), which can be decomposed as follows:

$$T(s_t, \theta_t, \mathcal{N}_t | s_{t-1}, a_{t-1}, \theta_{t-1}) = P_E(\theta_t | s_t, \mathcal{N}_t) P_O(s_t, \mathcal{N}_t | s_{t-1}, a_{t-1}, \theta_{t-1}), \quad (2)$$

where $P_O(s_t, \mathcal{N}_t | s_{t-1}, a_{t-1}, \theta_{t-1})$ describes a transition function generating a variable agent set $\mathcal{N}_t$ and a state $s_t$ that can be *observed* by each agent; the samples $\theta_t$ generated from $P_E(\theta_t | s_t, \mathcal{N}_t) = \prod_{j=1}^{|\mathcal{N}_t|} P_A(\theta_{t,j} | \{j\}, s_t)$ are *unobserved* by agents, however, it is critical to making decision, which motivates us to estimate this term in practice. An illustrative example to explain $P_O$ and $P_E$ is shown in Figure 4 in Appendix. $T$ in Eq. 2 overwrites the transition function in OSBG. $R_j : \mathcal{S} \times \mathcal{A}_{\mathcal{N}} \to \mathbb{R}_{\geq 0}$ extends agent $j$'s preference value of the stateless coalitional affinity games to state space and action space, named as agent $j$'s preference reward. In more details, $R_j(s_t, a_{t,\mathcal{C}})$ implies agent $j$'s preference reward of an arbitrary coalition $\mathcal{C} \subseteq \mathcal{N}_t$ with the corresponding coalitional action decided by $\pi_{t,\mathcal{C}} = \times_{j \in \mathcal{C}} \pi_j$. The affinity graph weight between agent $j$ and $j'$ is extended as $w_{jj'} : \mathcal{S} \times \mathcal{A}_j \times \mathcal{A}_{j'} \to \mathbb{R}$. Thereby, we can represent agent $j$'s preference reward for an arbitrary coalition $\mathcal{C}$ as $R_j(s_t, a_{t,\mathcal{C}}) = \sum_{(j,j') \in \mathcal{E}, j' \in \mathcal{C}} w_{jj'}(s_t, a_{t,j}, a_{t,j'})$.

**Proposition 1.** *The transition function $T(\theta_t, s_t, \mathcal{N}_t | \theta_{t-1}, s_{t-1}, a_{t-1})$ exists for $t \geq 1$ given that the following conditions hold: $P_T(\mathcal{N}_t, s_t | \mathcal{N}_{t-1}, s_{t-1}, a_{t-1})$ is well defined for $t \geq 1$; $P_I(\mathcal{N}_0 | s_0)$ is well defined; and $P_A(\theta_{t,j} | \{j\}, s_t)$ is well defined for $t \geq 0$.*

## 3.2 AD HOC TEAMWORK SETTING

In this subsection, we adapt OSB-CAG for the AHT setting in order to design algorithms for solving this problem. Under this setting, we can only control the learner $i$ to cooperate with other uncontrollable ad hoc teammates $-i$ with no prior coordination (Stone et al., 2010).

**Affinity Graph for Ad Hoc Teamwork.** We first discuss the affinity graph structure for the setting of ad hoc teamwork. Although the ad hoc teammates are assumed to have a common objective, they have no prior coordination, which motivates us to presume that there exists no relationship among these teammates (i.e., no edges needed when constructing an affinity graph). [1] On the other hand, the learner's goal is to establish collaboration (relationship) with the variable number of ad hoc teammates. To fulfil the above requirements we design the affinity graph as a star graph in Definition 2. Thereby, the preference reward of any teammate $j$ under the star graph is written as $R_j(s_t, a_t) = w_{ji}(s_t, a_{t,j}, a_{t,i})$ since it is only related to the learner $i$, while the learner's preference reward is written as $R_i(s_t, a_t) = \sum_{j \in -i} w_{ij}(s_t, a_{t,i}, a_{t,j})$.

**Definition 2.** *In the OSB-CAG with the ad hoc teamwork setting, the affinity graph is defined as a star graph where the learner is the internal node and the ad hoc teammates are leaves.*

**Ad Hoc Bayesian Strict Core.** To prevent the ad hoc teammates from leaving the grand coalition, we now extend strict core to AHT as a solution concept. Unlike the traditional CAG that focuses on finding a stable coalition structure given the preference values composed of predetermined affinity graph weights, OSB-CAG aims to maximize affinity graph weights via the learner's actions to stabilize the grand coalition $\mathcal{N}_t$ at every timestep $t$. To fit the setting of OSB-CAG, we replace $v_j$ by $R_j$, which naturally extends the strict core to OSB-CAG for any timestep $t$ with state $s_t$, agent set $\mathcal{N}_t$ and joint agent type $\theta_t$. Unfortunately, it is usually impossible to find the optimal weights fitting a task, without any relevant knowledge.

**Theorem 1.** *If a CAG is symmetric, then maximizing the social welfare under the grand coalition results in strict core stability.*

**Corollary 1.** *If the affinity graph in OSB-CAG is symmetric, then finding a joint policy $\pi_t$ that maximizes the social welfare $\sum_{j \in \mathcal{N}_t} R_j(s_t, a_t)$ for an arbitrary timestep $t$ with state $s_t$, agent set $\mathcal{N}_t$ and joint agent-type $\theta_t$ reaches strict core stability.*

For tractablility, we first prove a general result for CAG that the strict core stability can be reached by maximizing the social welfare as shown in Theorem 1. The realization of Theorem 1 for OSB-CAG at an arbitrary timestep $t$ is described in Corollary 1. Relaxing Corollary 1 to the long-term control with a stationary policy, we define ad hoc Bayesian strict core (AH-BSC) focusing on finding the learner's optimal stationary policy to maximize the social welfare under any grand coalition $\mathcal{N}_t \subseteq \mathcal{N}$ at any timestep $t$ in the long term such that

$$\text{AH-BSC} := \left\{ \pi_i^* \,\middle|\, \forall s_0 \in \mathcal{S}, \forall \pi_i, \, \mathbb{E}_{\pi_i^*}\Big[ \sum_{t=0}^{\infty} \gamma^t \sum_{j \in \mathcal{N}_t} R_j(s_t, a_t) \Big] \geq \mathbb{E}_{\pi_i}\Big[ \sum_{t=0}^{\infty} \gamma^t \sum_{j \in \mathcal{N}_t} R_j(s_t, a_t) \Big] \right\}, \quad (3)$$

---

[1] Note that we do not consider the cases where ad hoc teammates are adaptive to establish connection with others like the learner, which would lead to the problem more complicated, even with no solution.

where $\mathbb{E}[\cdot]$ denotes the expectation over $\theta_t \sim P_E, \mathcal{N}_t, s_t \sim P_O, a_{t,-i} \sim \pi_{t,-i}$ and $a_{t,i} \sim \pi_i$.

# 4 COALITIONAL AFFINITY GPL

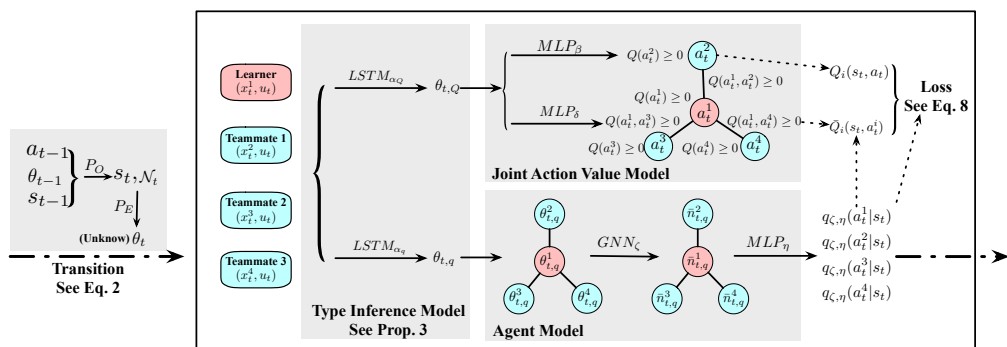

Figure 1: Illustration of the proposed algorithm named as CA-GPL, where Proposition 3 is shown in Appendix C due to the page limit.

In this section, we investigate how AH-BSC can be formulated as an RL problem (see Section 4.1), how the preference Q-values indicating each agent's long-term preference can be represented (see Section 4.2), and how the RL problem can be solved in theory (see Section 4.3) and in practice (see Section 4.4). Combining all above results, we propose an algorithm named as coalitional affinity GPL (CA-GPL), whose procedure is summarized in Figure 1.

## 4.1 SOLVING AD HOC BAYESIAN STRICT CORE AS A REINFORCEMENT LEARNING PROBLEM

At the beginning, let us presume that a global reward that the learner aims to maximize in the long term to encourage collaboration can be equivalently defined as the social welfare concerning a coalition structure $\mathcal{CS}_t$ such that $R(s_t, a_t) = \sum_{\mathcal{C} \in \mathcal{CS}_t} \sum_{j \in \mathcal{C}} R_j(s_t, a_{t,c})$. Note that this interpretation is without loss of generality because it considers all agents dedicated to the environment. In this work, we aim at finding a deterministic optimal policy for the learner, so that Eq. 3 agrees with the condition of optimal learner's policy for OSBG, but from a completely different view. As a result, we can maximize the cumulative global rewards under the grand coalition (i.e. $\mathcal{CS}_t = \{\mathcal{N}_t\}$) to implement the AH-BSC via the learner's policy $\pi_i$ such that

$$\max_{\pi_i} \mathbb{E}_{\theta_t \sim P_E,\, a_{t,-i} \sim \pi_{t,-i},\, a_{t,i} \sim \pi_i,\, \mathcal{N}_t, s_t \sim P_O} \Big[ \sum_{t=0}^{\infty} \gamma^t R(s_t, a_t) \Big]. \tag{4}$$

## 4.2 REPRESENTATION OF PREFERENCE Q-VALUES

Each agent $j$'s preference Q-value is the expectation of its discounted cumulative preference rewards such that $Q_j(s_t, a_t) = \mathbb{E}[\sum_{t=0}^{\infty} \gamma^t R_j(s_t, a_t)]$. For a teammate $j$, its preference reward of the grand coalition is defined as $R_j(s_t, a_t) = w_{ji}(s_t, a_{t,j}, a_{t,i})$, while for a learner $i$, its preference reward of the grand coalition is defined as $R_i(s_t, a_t) = \sum_{j \in -i} w_{ij}(s_t, a_{t,i}, a_{t,j})$. Next, we prove that $w_{jj'} \geq 0$ for every agent $j$ promises the strict core stability of the grand coalition as Theorem 2 shows. This condition can be seen as an inductive bias to be considered in representation of the value function, so that the search space of strict core stability following Eq. 4 would be well confined.

**Theorem 2.** *In an OSB-CAG for ad hoc teamwork, if for all $(j, j') \in \mathcal{E}$, all states $s \in \mathcal{S}$ and all action pairs $(a_j, a_{j'}) \in \mathcal{A}_j \times \mathcal{A}_{j'}$, $w_{jj'}(s, a_j, a_{j'}) \geq 0$, then the grand coalition exhibits strict core stability.*

On the other hand, the individual rationality following the convention of the CAG (see Section 2.1) needs to be guaranteed such that for an agent $j'$, $R_{j'}(s_t, a_t) \geq R_{j'}(s_t, a_{t,j'})$, where $R_{j'}(s_t, a_{t,j'})$ is defined as the preference reward when only an agent $j'$ itself forms a singleton coalition. Besides, the symmetry of affinity graph weights mentioned in Theorem 1 under the star affinity graph also needs to be guaranteed. Without loss of generality, we define the affinity graph weights in Definition 3, based on $\alpha_{ij}, \alpha_{ji}, R_j(s_t, a_{t,j})$ and $R_i(s_t, a_{t,i})$, whose designs will be discussed in the next paragraph.

**Definition 3.** *Given the definition of $\alpha_{ij}$, $\alpha_{ji}$, $R_j(s_t, a_{t,j})$ and $R_i(s_t, a_{t,i})$, the affinity graph weights are defined as follows: $w_{ji}(s_t, a_{t,j}, a_{t,i}) = \alpha_{ji}(s_t, a_{t,j}, a_{t,i}) + \frac{1}{2} \cdot R_j(s_t, a_{t,j}) + \frac{1}{2|-i|} \cdot R_i(s_t, a_{t,i})$ and $w_{ij}(s_t, a_{t,i}, a_{t,j}) = \alpha_{ij}(s_t, a_{t,i}, a_{t,j}) + \frac{1}{2|-i|} \cdot R_i(s_t, a_{t,i}) + \frac{1}{2} \cdot R_j(s_t, a_{t,j})$.*

Suppose that $R_j(s_t, a_{t,j}) = \frac{1}{|-i|} \cdot R_i(s_t, a_{t,i})$ at timestep $t$, we prove in Proposition 2 that if $\alpha_{ij}(s_t, a_{t,i}, a_{t,j}) = \alpha_{ji}(s_t, a_{t,j}, a_{t,i}) \geq 0$, $R_j(s_t, a_{t,j}) \geq 0$ and $R_i(s_t, a_{t,i}) \geq 0$, then the individual rationality, the symmetric affinity graph weights and the grand coalition exhibiting strict core stability would be simultaneously satisfied.

**Proposition 2.** *For the learner $i$ and any teammate $j$ at timestep $t$, suppose that $R_j(s_t, a_{t,j}) = \frac{1}{|-i|} \cdot R_i(s_t, a_{t,i})$ and the affinity graph weights of a star graph are defined as Definition 3 shows, $\alpha_{ij} = \alpha_{ji} \geq 0$, $R_j \geq 0$ and $R_i \geq 0$ would lead to symmetry of affinity weights, individual rationality and the grand coalition exhibiting strict core stability.*

Considering the long-term goal and symmetry, we obtain the following two terms by Lemma 1 [2] such that $Q_{ji}(s_t, a_{t,j}, a_{t,i}) = Q_{ij}(s_t, a_{t,i}, a_{t,j}) = \mathbb{E}[\sum_{t=0}^{\infty} \gamma^t \alpha_{ij}(s, a_{t,i}, a_{t,j})]$ for any teammate $j$ and $Q_{j'}(s_t, a_{t,j'}) = \mathbb{E}[\sum_{t=0}^{\infty} \gamma^t R_{j'}(s_t, a_{t,j'})]$ for any agent $j'$. Due to the linearity of expectation and the definition of preference Q-values, a teammate's preference Q-value is defined as that $Q_j(s_t, a_t) = Q_{ji}(s_t, a_{t,j}, a_{t,i}) + \frac{1}{2} \cdot Q_j(s_t, a_{t,j}) + \frac{1}{2|-i|} \cdot Q_i(s_t, a_{t,i})$, whereas the learner's preference Q-value is defined as that $Q_i(s_t, a_t) = \sum_{j \in -i} Q_{ij}(s_t, a_{t,i}, a_{t,j}) + \frac{1}{2} \cdot \sum_{j \in -i} Q_j(s_t, a_{t,j}) + \frac{1}{2} \cdot Q_i(s_t, a_{t,i})$. A necessary condition to satisfy the condition in Proposition 2 is that $Q_{ij} = Q_{ji} \geq 0$, $Q_i \geq 0$ and $Q_j \geq 0$.

**Lemma 1.** *Suppose that $\alpha_{jj'}(s_t, a_{t,j}, a_{t,j'}) = 0$ for $t \geq T$ if agent $j$ or $j'$ leaves the environment at timestep $T$ and $R_j(s_t, a_{t,j}) = 0$ for $t \geq T$ if agent $j$ leaves the environment at timestep $T$, then $Q_{jj'}(s_t, a_{t,j}, a_{t,j'}) = \mathbb{E}[\sum_{t=0}^{\infty} \gamma^t \alpha_{jj'}(s, a_{t,j}, a_{t,j'})]$ and $Q_j(s_t, a_{t,j}) = \mathbb{E}[\sum_{t=0}^{\infty} \gamma^t R_j(s_t, a_{t,j})]$.*

### 4.3 OPEN AD HOC BELLMAN OPTIMALITY EQUATION

Akin to the Bellman optimality equation in RL, by Theorem 3 (i.e., proved based on Lemma 1) we propose the open ad hoc Bellman optimality equation (OAH-BOE) to describe the condition of the learner's optimal policy such that

$$Q^*(s_t, a_t) = R(s_t, a_t) + \gamma \mathbb{E}_{\mathcal{N}_{t+1}, s_{t+1} \sim P_O}\Big[ \max_{a_i'} \mathbb{E}_{\substack{\theta_{t+1} \sim P_E, \\ a_{t+1,-i} \sim \pi_{t+1,-i}}} \Big[ Q^*(s_{t+1}, a_{t+1,-i}, a_i') \Big] + b(s_{t+1}) \Big], \quad (5)$$

where $Q^*(s_t, a_t) = \sum_{j \in \mathcal{N}_t} Q_j^*(s_t, a_t)$ and $b(s_{t+1}) \geq 0$. It is undefined when $\mathcal{N}_t \subset \mathcal{N}_{t+1}$, since it is unreasonable to consider an agent $j \in \mathcal{N}_{t+1}$ but $\notin \mathcal{N}_t$ at timestep $t$ in deriving Bellman optimality equation, which is problematic.

**Theorem 3.** *Suppose that $\alpha_{jj'}(s_t, a_{t,j}, a_{t,j'}) = 0$ for $t \geq T$ if agent $j$ or $j'$ leaves the environment at timestep $T$ and $R_j(s_t, a_{t,j}) = 0$ for $t \geq T$ if agent $j$ leaves the environment at timestep $T$, then the open ad hoc Bellman optimality equation is defined as Eq. 5.*

To solve the OAH-BOE, we further propose the following operator $\Gamma : Q \mapsto \Gamma Q$ analogous to the Bellman operator named as open ad hoc Bellman operator (OAH-BO) such that

$$\Gamma Q(s_{t+1}, a_{t+1,-i}, a_i') = R(s_t, a_t) + \gamma \mathbb{E}_{P_O}\Big[ \max_{a_i'} \mathbb{E}_{P_E, \pi_{t+1,-i}}[Q(s_{t+1}, a_{t+1,-i}, a_i')] + b(s_{t+1}) \Big]. \quad (6)$$

Similar to OAH-BOE, it is undefined when $\mathcal{N}_t \subset \mathcal{N}_{t+1}$. If we define a new global reward function such that $\hat{R}(s_t, a_t) = R(s_t, a_t) + \gamma \mathbb{E}_{P_O}[b(s_{t+1})]$, then the OAH-BO is in the same form as the normal Bellman operator if $P_E$ and $\pi_{t+1,-i}$ are known in prior. Therefore, recursively running Eq. 6 converges to the OAH-BOE, following the well-known value iteration convergence results (Sutton & Barto, 2018). By stochastic approximation, we can derive the relevant Q-learning algorithm (Watkins & Dayan, 1992) and its fitted Q-learning (Ernst et al., 2005) objective function to deal with continuous or very large discrete states and/or action spaces in practice as follows:

$$\min_{\beta} \mathbb{E}\Big[ \frac{1}{2} \Big( R(s_t, a_t) + \gamma \Big( \max_{a_i'} \mathbb{E}_{P_E, \pi_{t+1,-i}}[Q^-(s_{t+1}, a_{t+1,-i}, a_i')] + b(s_{t+1}) \Big) - Q(s_t, a_{t,-i}, a_i; \beta) \Big)^2 \Big],$$
$$(7)$$

where $Q^-$ is the target global Q-value and $Q$ is assumed to be parameterised by $\beta$.

---

[2] The insight behind Lemma 1 is shown in Remark 2 in Appendix D.

### 4.4 PRACTICAL IMPLEMENTATION

As our theoretical model defines, a global reward needs to be greater than or equal to zero, while the global reward of an actual environment $R(s_t, a_t)$ could be less than zero. This can be adjusted by adding the maximum difference among all states and joint actions between these two global reward functions denoted as $\Delta R$ to $R(s_t, a_t)$ to construct a surrogate global reward function $\hat{R}(s_t, a_t)$ without changing the original goal such that $\hat{R}(s_t, a_t) = R(s_t, a_t) + \Delta R$. Since $\delta = \mathbb{E}[\Delta R + \gamma b(s_{t+1})]$ is a constant term, it can be seen as a bias which results in the minimum of the objective function in Eq. 7 changing from $0$ to $0.5\delta^2$, but with the same optimal solution. For this reason, we can ignore the bias term $\delta$, and therefore the objective function becomes:

$$\min_{\beta} \mathbb{E}\Big[\frac{1}{2}\Big(R(s_t, a_t) + \gamma \max_{a'_i} \mathbb{E}_{P_E, \pi_{t+1,-i}}[Q^-(s_{t+1}, a_{t+1,-i}, a'_i)] - Q(s_t, a_{t,-i}, a'_i; \beta)\Big)^2\Big]. \quad (8)$$

Eq. 8 is further solved by the deep Q-learning framework (DQN) (Mnih et al., 2013) in practice. As for the architecture, our method can be easily implemented based on GPL framework (Rahman et al., 2021), whose details are mentioned in Appendix F. More specifically, the coordination graph in GPL is configured to be a star graph. To satisfy the conditions discussed in Proposition 2, we need only to learn $Q_{ij}$ rather than both $Q_{ij}$ and $Q_{ji}$ to satisfy symmetry, as well as regulate $Q_i \geq 0$, $Q_j \geq 0$ and $Q_{ij} \geq 0$ for any teammate $j$ by adding the abs function to the outputs. For this reason, we name our algorithm coalitional affinity GPL (CA-GPL) to credit the introduction of CAG. Please refer to Appendix C for more details about the deeper understanding of the GPL framework using our theory.

**Remark 1.** *In practice, we implement CA-GPL following the GPL framework where it does not ignore the transitions of $\mathcal{N}_t \subset \mathcal{N}_{t+1}$ during training (violating the condition of Eq. 6 and 7, and therefore Eq. 8). Nevertheless, this only emerges at the timestep when a new member joins the team, which is a small fraction of data among all. In other words, there is only a small fraction of pathological updates, which would not affect validity of the learning progress too much.*

## 5 EXPERIMENTS

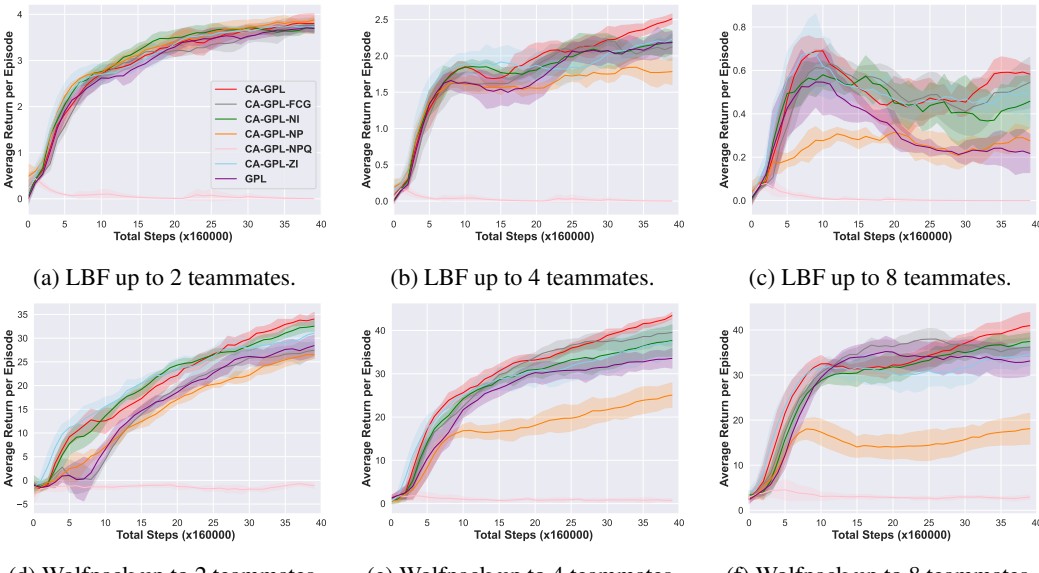

(a) LBF up to 2 teammates.  (b) LBF up to 4 teammates.  (c) LBF up to 8 teammates.

(d) Wolfpack up to 2 teammates.  (e) Wolfpack up to 4 teammates.  (f) Wolfpack up to 8 teammates.

Figure 2: Evaluation on different environments with variant upper limit numbers of teammates. All figures share one legend in Fig. 2a. Our proposed method, CA-GPL, surpasses the state-of-the-art GPL and our variants (CA-GPL-×) in every scenario, except for LBF with up to 2 teammates, where the performance of all methods is almost the same.

We evaluate the proposed CA-GPL in two existing environments, LBF and Wolfpack, with open team settings (Rahman et al., 2021; 2022), where teammates are randomly selected to enter the environment

and stay for a certain number of time steps. In experiments, we train the learner in an environment where at most 2 teammates exist at each timestep, while we test the learner in environments where at most 2, 4, and 8 teammates exist at each timestep respectively to show the ability to tackle both unseen compositions and team sizes. All experiments are carried out with five random seeds, and the results are demonstrated as the mean performance with a 95% confidence interval. We design experiments to mainly verify the following statements such that (1) the GPL settings (to form CA-GPL) derived from the theoretical model are effective in improving the performance; (2) the theoretical model is reasonable so that the affinity graph weights can be aligned with the cooperative behaviors. The details of experimental setups are described in Appendix G.

**Baselines.** The baselines we use in this experiment are GPL-Q (shortened as GPL) (Rahman et al., 2021) and variants of the proposed CA-GPL to validate the effectiveness of the theoretical model: **CA-GPL-NI** and **CA-GPL-NP** are variants that enforce negative individual utility terms and pairwise utility terms, i.e., $R_{j'} < 0$ and $R_{ij} < 0$, respectively; **CA-GPL-NPQ** refers to enforcing negative both individual utility terms and pairwise utility terms; **CA-GPL-ZI** is a variant with zero as individual utility terms implying the complete ignorance of individual rationality; **CA-GPL-FCG** replaces the star graph with a fully connected graph with other settings that are still valid to guarantee our theoretical claims (see Proposition 4 in Appendix E).

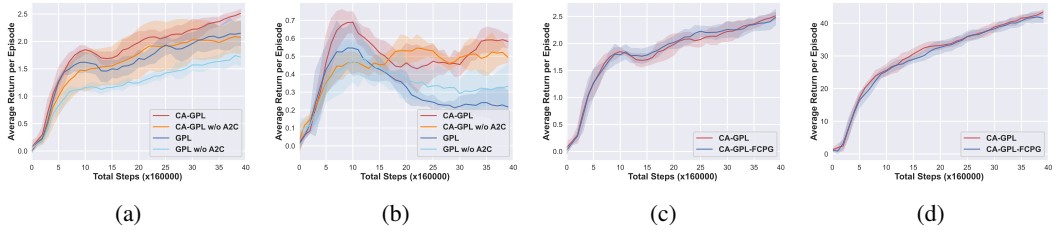

Figure 3: *Fig. 3a and Fig. 3b*: Stability of performance comparison in two different sets of teammate type. The suffix "w/o A2C" denotes the "teammate type set without pre-trained A2C agent". Fig. 3a shows the results on the maximum teammate size of 4, and Fig. 3b for up to 8 on LBF. *Fig. 3c and Fig. 3d*: Comparison between different graph structures for GNNs to aggregate observations for teammates' policies in LBF (Fig. 3c) and Wolfpack (Fig. 3d) environments with up to 4 teammates.

## 5.1 MAIN RESULTS

We first show the main results of three test environments where up to 2, 4 and 8 ad hoc teammates exist for LBF and Wolfpack, respectively. As Fig. 2 shows, CA-GPL performs generally best among all variants, especially exceeding GPL with a large margin in most scenarios. The better performance compared to each CA-GPL variant in most scenarios shows the effectiveness of every component mentioned in the proposed theoretical model. Among all these CA-GPL variants, it is noticeable that CA-GPL-NI and CA-GPL-FCG also generally perform well. The most probable reason for the comparatively good performance of CA-GPL-NI is that $R_{j'} \geq 0$ is only a sufficient condition to satisfy the strict core stability and individual rationality, which implies that there may exist some other feasible conditions. Due to the lack of exact constraints, it is not easy to reach these conditions during the learning. The comparatively good performance of CA-GPL-FCG could be due to that there still exists relationship among ad hoc teammates in some environments, but not strong enough, so its performance is still worse than CA-GPL.

## 5.2 EVALUATION ON DIFFERENT TEAMMATE SETS AND GRAPH STRUCTURES

**Sensitivity to Different Teammates.** We observe from experimental results (see Fig. 3a and 3b) as well as previous results (Rahman et al., 2021) that the learner's performance is highly dependent on different teammate types. For this reason, we introduce a novel case study to investigate the stability of performance for different sets of teammate types between CA-GPL and GPL. For simplicity, we construct two teammate type sets such that one is with a pretrained A2C agent and another is without. Each teammate type set is used for training and evaluation, rather than training with a teammate type set but testing with another one in previous works (Rahman et al., 2021). As depicted in Fig. 3a and

Fig. 3b, a comparative analysis was conducted on the average return in two distinct sets of teammates, i.e., w/ and w/o A2C agent pre-trained. In both LBF settings, adapting to up to 4 teammates (Fig. 3a) and extending up to 8 teammates (Fig. 3b), the discrepancy of performances between the two type sets by our CA-GPL method is considerably less pronounced in comparison with GPL. Furthermore, we conducted a qualitative analysis to assess performance stability, as shown in Table. 1. We first employ a metric such that $(|\mathcal{R}_{w/} - \mathcal{R}_{w/o}|)/\mathcal{R}_{w/o} \times 100$ to quantify the variance of performances between two distinct sets of teammate type, where $\mathcal{R}$ is defined as the average episodic return. The lower the value, the more is the stability. The results of the latter half of the training phase on LBF with two team configurations substantiate that CA-GPL exhibits superior stability in terms of both the period value and the average value.

Table 1: Qualitative stability assessment during the latter half of the training phase across varied teammate type sets within the LBF environment. The lower is the value, the more stable is the result.

| Team Size | Method | Training Interval | | | | | Avg. |
|---|---|---|---|---|---|---|---|
| | | 50%-60% | 60%-70% | 70%-80% | 80%-90% | 90%-100% | |
| Up to 4 Teammates | GPL | 37.46 | 38.99 | 25.33 | 31.12 | 26.30 | 31.84 ± 5.59 |
| | CA-GPL | 7.33 | 8.41 | 8.08 | 21.90 | 19.37 | **13.02 ± 6.28** |
| Up to 8 Teammates | GPL | 32.32 | 22.76 | 29.46 | 26.07 | 38.58 | 29.84 ± 5.42 |
| | CA-GPL | 14.79 | 4.05 | 5.09 | 22.48 | 8.92 | **11.07 ± 6.84** |

**Graph Structure to Aggregate Observations for Teammates' Policies.** In GPL, the graph structure for GNNs to aggregate others' observations for each teammate's policy is defined as a fully connected graph (see Appendix F). In other words, each teammate's policy model receives observations from all other agents. Although our theoretical model is irrelevant to teammates' policies, we employ the star graph for GNNs to aggregate observations for teammates' policies to preserve the consistency of the affinity graph in the CA-GPL. Under this scheme, each teammate's policy model only receives the learner's observation. To verify whether the above scheme is enough to model teammates' policies, we compare CA-GPL with its variant with a fully connected graph for aggregating observations for teammates' policies named CA-GPL-FCPG. As Fig. 3c and 3d show, CA-GPL and CA-GPL-FCPG possess similar performances on Wolfpack and LBF with up to 4 teammates. The same phenomenon also appears in environments with up to 8 teammates (see Appendix H). The probable reason could be that the relationships among teammates are weak so that information from the agents other than the learner is unable to influence a teammate's decision-making, which verifies and consolidates modeling the coordination graph as a star graph.

## 6 CONCLUSION

**Summary.** This work proposes a theoretical model from the perspective of cooperative game theory to address a challenging problem called open ad hoc teamwork. In comparison with prior works on modelling this problem as a non-cooperative game theoretical model Our theoretical model can better establish the learner's influence on ad hoc teammates so that the open ad hoc teamwork process is well described. Based on the theoretical model, we also propose a novel algorithm named CA-GPL, which can be easily implemented based on the SOTA algorithm GPL with some theoretically guaranteed settings from this work. To this end, it also provides an interpretation of the heuristic implementation of GPL. In experiments, we demonstrate and verify the effectiveness and reasonableness of our theoretical model through delicate experimental designs and visualizations.

**Future Work.** This work is the first work establishing a theoretical model and a practical algorithm standing by the view of cooperative game theory, which sheds light on incorporating cooperative game theory into ad hoc teamwork. Multiple future directions can be taken after this work. First, to simplify the problem, we have ignored the relationship among teammates. However, in some realistic scenarios, we believe this relationship is critical to the learner's decision. One can refer to the concept of top responsiveness (Alcalde & Revilla, 2004) to gain more insight into this direction. Another possible direction is how our theoretical model can be generalized to the scenarios with teammates and enemies. Regarding this direction, one can refer to the concept of friends and enemies (Dimitrov et al., 2006) as well as B- and W-preferences (Cechlárová & Romero-Medina, 2001; Cechlárová & Hajduková, 2003) in hedonic games.

ETHICS STATEMENT

Our method and algorithm do not involve any adversarial attack, and will not endanger human security. All our experiments are performed in the simulation environment, which does not involve ethical and fair issues.

REPRODUCIBILITY STATEMENT

The source code of this paper will be published later. We specify all the experiment implementation details, the experiment setup, and the additional results in Appendix G, Appendix H, and our project website https://sites.google.com/view/cagpl/.

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

# A RELATED WORK

**Theoretical Models for Ad Hoc Teamwork.** We first discuss the previous works on the theoretical models used to describe the AHT. Brafman & Tennenholtz (1996) was the first to investigate ad hoc teamwork in theory by studying the repeated matrix game with a single teammate. Agmon & Stone (2012) extended this thread to multiple teammates, while Agmon et al. (2014) further relaxed the known teammates' policies to the policies drawn from a known set. Stone & Kraus (2010) proposed collaborative multi-armed bandits, which initially formalized the AHT but with strong assumptions (e.g. teammates' policies and environments were known to the ad hoc agent). The first work proposed as a complete theoretical model to solve dynamic environments and unknown teammates in the AHT by combining Markov and Bayesian games is a stochastic Bayesian game (SBG) (Albrecht & Ramamoorthy, 2013). Extended from the SBG, a theoretical model called open stochastic Bayesian game (OSBG) (Rahman et al., 2021) was proposed to address the OAHT. Zintgraf et al. (2021) modelled the AHT as interactive Bayesian reinforcement learning (IBRL) in Markov games, whose purpose was to solve non-stationary teammates' policies during one episode but not across episodes. Distinguished from the above purpose, Xie et al. (2021) proposed to use a hidden parameter Markov decision process (HiP-MDP) to solve the situation where teammates' policies vary across episodes but maintain stationary during each episode. In this paper, we formalize the AHT by the CAG (a.k.a. additively separable games) from the view of cooperative game theory as well as propose a theoretical model named OSB-CAG and the solution concept AH-BSC to characterize better the interactive process of ad hoc teamwork (i.e., the influences between the learner and the ad hoc teammates to encourage collaboration).

**Algorithms for Ad Hoc Teamwork.** We now look into AHT from the algorithmic perspective. The best response algorithm was proposed to solve the problem under assumptions of a matrix game and the well-known teammates' policies (Stone et al., 2009). Extending this line of research, REACT (Agmon et al., 2014) was proposed that works well on many matrices where teammates' policies are drawn from a known set. Wu et al. (2011) used biased adaptive play to estimate teammates' actions based on their history actions and Monte Carlo tree search to plan the ad hoc agent's actions. HBA (Albrecht & Ramamoorthy, 2013) was proposed to solve the problems beyond matrix games, which maintain a probability distribution of predetermined agent types and maximize the long-term payoffs by an extended Bellman operator. PLASTIC-Policy (Barrett et al., 2017) was proposed to solve more realistic scenarios like RoboCup (Kalyanakrishnan et al., 2007) that trains teammates' policies by behaviour cloning and the ad hoc agent's policy w.r.t. the state and teammates' types by FQI (Ernst et al., 2005). During the test, the belief of agent types is updated via the pretrained teammates' policies and the observed teammates' reactions, which is used to estimate teammates' types and their actions to select the ad hoc agent's action. AATEAM (Chen et al., 2020) extended PLASTIC-Policy with the attention network (Bahdanau et al., 2014) for improving the estimation of unseen agent types. Rahman et al. (2021) extended HBA with modern deep learning techniques such as GNNs and RL algorithms to solve the OAHT and proposed GPL. Gu et al. (2022) proposed ODITS to tackle teammates with rapidly changing behaviors under partial observability. In this paper, we define the affinity graph as a star graph and derive the representation of the global Q-value. As per the theoretical model and the derived settings, we propose an interpretable algorithm named CA-GPL that can be easily implemented based on GPL.

**Relationship to Cooperative Multi-Agent Reinforcement Learning.** Cooperative MARL primarily aims at training and controlling agents altogether to optimally complete a common-goal task. The popular research topics are value decomposition/credit assignment (Foerster et al., 2018; Sunehag et al., 2018; Rashid et al., 2018), reward shaping (Du et al., 2019; Mguni et al., 2022), communication (Foerster et al., 2016; Sukhbaatar et al., 2016; Jiang & Lu, 2018; Kim et al., 2019), and etc.. In this paper, only one agent is trained to interact with an unknown set of uncontrollable agents to complete a common-goal task. For this reason, the joint policy in the AHT could be sub-optimal in comparison with that in the MARL. To solve value decomposition/credit assignment, the transferable utility games in cooperative game theory was introduced and extended to MARL for the application of Shapley value (Wang et al., 2020; 2022). In this paper, we introduce the CAG that belongs to non-transferable utility games (i.e. a broader class involving transferable utility games) to solve the OAHT. Recently, some works start to utilize causal analysis to analyze multi-agent systems (Hammond et al., 2023) and improve training MARL (Li et al., 2022). Correspondingly, we apply causal inference in this paper to derive a Bayesian filter to approximate and estimate the joint type assignment function.

# B    ASSUMPTIONS

**Assumption 1.** *There exists an underlying agent type set to generate ad hoc teammates in an environment which is unknown to the learner.*

**Assumption 2.** *The ad hoc teammates can be influenced by the learner through its decision making.*

**Assumption 3.** *The agents stay in the environment at least for a period of timesteps.*

**Assumption 4.** *Each teammate of an arbitrary agent type is equipped with a fixed policy.*

Assumption 1 is a natural way to describe agent types of teammates. If the agent type set is large enough, it is impossible to encounter all agent types or the compositions of agent types. For this reason, this assumption would not corrupt the generalizability of the open ad hoc teamwork.

Assumption 2 is a fundamental and common property to rationalize the ad hoc teamwork problem. It is usually described as the reactivity of teammates (Barrett et al., 2017). If all ad hoc teammates cannot react to or be influenced by the learner, then the problem would be degraded to a single agent problem with teammates as moving "obstacles". To avoid such a pathological situation, we would maintain this assumption as a boundary of ad hoc teamwork.

Assumption 3 is a condition to guarantee the possibility to complete an arbitrary task. If an agent joins at some timestep and leaves instantaneously at the next timestep, there is almost no chance for any team of agents to react to and influence each other.

Assumption 4 is a condition to simplify the analysis of convergence to the optimal policy of the learner. More specifically, to the learner, the condition of teammates' fixed policies enables the Markov process stationary.

# C    FURTHER DETAILS OF IMPLEMENTATION

Since $P_E$ and $\pi_{t,-i}$ are unknown to the learner, we need to discuss how to estimate these two terms to enable convergence of Eq. 6. A Bayesian filter $\hat{P}_E(\theta_t|s_t, \theta_{t-1}^-, \theta_{t-1}^+) = \prod_{j=1}^{|\mathcal{N}_t|} \hat{P}_A(\theta_{t,j}|\theta_{t-1,j}, s_t)$ is derived by Proposition 3 for the learner to generate its belief in the types of teammates. If agent $j$ is a newly joined agent at timestep $t$, $\theta_{t-1,j}$ would be zeros for $\hat{P}_A(\theta_{t,j}|\theta_{t-1,j}, s_t)$, otherwise, it would be the estimate for timestep $t-1$. On the other hand, a Bayesian filter is also used to infer the belief of states via the agents' joint observation history (Oliehoek & Amato, 2016), which is assumed to be factorized in agents like $\hat{P}_A$. In other words, such an aggregated Bayesian filter for each agent takes as input its local observation and its belief of the aggregation its types and locally inferred state for the last timestep and generates its belief for the current timestep as output. As a result, we use an LSTM (Hochreiter & Schmidhuber, 1997) to approximate the aggregated Bayesian filter.

**Proposition 3.** $P_E(\theta_t|\mathcal{N}_t, s_t)$ *can be approximated as the expression* $\hat{P}_E(\theta_t|s_t, \theta_{t-1}^-, \theta_{t-1}^+)$, *where* $\theta_{t-1}^-$ *indicates the types of remaining agents of* $\mathcal{N}_{t-1}$ *and* $\theta_{t-1}^+$ *indicates the unknown types of newly joined agents at timestep* $t$ *(set to zeros as default).*

Furthermore, $\hat{\pi}_{t,-i} = \times_{j\in-i}\hat{\pi}_{t,j}$ is fitted by supervised learning through the behavior of the teammates during the interactions. The architecture of $\hat{\pi}_{t,j}$ is a shared graph neural network (GNN) aggregating each agent's belief for the current timestep, connected by the exclusive multilayer perceptrons (MLPs) to output teammate $j$'s actions. All $Q_i$, $Q_j$ and $Q_{ij}$ are approximated by MLPs, whose inputs are also generated by the approximate aggregated Bayesian filter as an LSTM.

# D    PROOFS

**Proposition 1.** *The transition function* $T(\theta_t, s_t, \mathcal{N}_t|\theta_{t-1}, s_{t-1}, a_{t-1})$ *exists for* $t \geq 1$ *given that the following conditions hold:* $P_T(\mathcal{N}_t, s_t|\mathcal{N}_{t-1}, s_{t-1}, a_{t-1})$ *is well defined for* $t \geq 1$; $P_I(\mathcal{N}_0|s_0)$ *is well defined; and* $P_A(\theta_{t,j}|\{j\}, s_t)$ *is well defined for* $t \geq 0$.

*Proof.* To ease the analysis and description, we assume that $s_t$ and $a_t$ are discrete with no loss of generality. We prove the existence of $T(\theta_t, s_t, \mathcal{N}_t | \theta_{t-1}, s_{t-1}, a_{t-1})$ as follows:

$$
\begin{aligned}
&T(\theta_t, s_t, \mathcal{N}_t | \theta_{t-1}, s_{t-1}, a_{t-1}) \\
&= P(\theta_t | \mathcal{N}_t, s_t, \theta_{t-1}, s_{t-1}, a_{t-1}) P_O(\mathcal{N}_t, s_t | \theta_{t-1}, s_{t-1}, a_{t-1}) \\
&= P_E(\theta_t | \mathcal{N}_t, s_t) P_O(\mathcal{N}_t, s_t | \theta_{t-1}, s_{t-1}, a_{t-1}) \text{ (By conditional independence)} \\
&= P_E(\theta_t | \mathcal{N}_t, s_t) \sum_{\mathcal{N}_{t-1}} P(\mathcal{N}_t, s_t, \mathcal{N}_{t-1} | \theta_{t-1}, s_{t-1}, a_{t-1}) \\
&= P_E(\theta_t | \mathcal{N}_t, s_t) \sum_{\mathcal{N}_{t-1}} P_T(\mathcal{N}_t, s_t | \theta_{t-1}, \mathcal{N}_{t-1}, s_{t-1}, a_{t-1}) P(\mathcal{N}_{t-1} | \theta_{t-1}, s_{t-1}, a_{t-1}) \\
&= P_E(\theta_t | \mathcal{N}_t, s_t) \sum_{\mathcal{N}_{t-1}} P_T(\mathcal{N}_t, s_t | \mathcal{N}_{t-1}, s_{t-1}, a_{t-1}) P(\mathcal{N}_{t-1} | \theta_{t-1}, s_{t-1}) \\
&= \prod_{j=1}^{|\mathcal{N}_t|} P_A(\theta_{t,j} | \{j\}, s_t) \sum_{\mathcal{N}_{t-1}} P_T(\mathcal{N}_t, s_t | \mathcal{N}_{t-1}, s_{t-1}, a_{t-1}) P(\mathcal{N}_{t-1} | \theta_{t-1}, s_{t-1}) \\
&\quad\quad\quad\quad\quad\quad\quad\quad\quad\quad\quad\quad\quad\quad\quad\quad\quad \text{(By conditional independence)}.
\end{aligned}
$$

We next show the existence of $P(\mathcal{N}_t | \theta_t, s_t)$ as follows:

$$
P(\mathcal{N}_t | \theta_t, s_t) = \frac{\sum_{s_t} P_E(\theta_t | \mathcal{N}_t, s_t) P(\mathcal{N}_t, s_t)}{\sum_{\mathcal{N}_t} \sum_{s_t} P_E(\theta_t | \mathcal{N}_t, s_t) P(\mathcal{N}_t, s_t)}.
$$

We now start to prove the existence of $P(\mathcal{N}_t, s_t)$ using mathematical induction as follows:

*Base case*: As per the definition, $P_I(\mathcal{N}_0, s_0)$ exists for $t = 0$.

*Induction case*: Assume the induction hypothesis that $P(\mathcal{N}_t, s_t)$ exists for any $t \geq 0$. Next, we prove the existence of $P(\mathcal{N}_{t+1}, s_{t+1})$ based on the induction hypothesis such that

$$
P(\mathcal{N}_{t+1}, s_{t+1}) = \sum_{\mathcal{N}_t} \sum_{s_t} \sum_{a_t} P(\mathcal{N}_{t+1}, s_{t+1}, \mathcal{N}_t, s_t, a_t).
$$

Since

$$
P(\mathcal{N}_{t+1}, s_{t+1}, \mathcal{N}_t, s_t, a_t) = P(\mathcal{N}_{t+1}, s_{t+1} | \mathcal{N}_t, s_t, a_t) P(\mathcal{N}_t, s_t) \pi(a_t | s_t),
$$

we have proved the existence of $P(\mathcal{N}_{t+1}, s_{t+1})$.

*Conclusion*: $P(\mathcal{N}_t, s_t)$ exists for any $t \geq 0$. $\quad\square$

**Lemma 2** (Brânzei & Larson (2009))**.** *If a CAG is symmetric, then the social-welfare maximizing partition exhibits inner stability.*

**Theorem 1.** *If a CAG is symmetric and the grand coalition is assumed to be strict core stable, then maximizing the social welfare under the grand coalition results in strict core stability.*

*Proof.* It is not difficult to observe that if the grand coalition is assumed to be strict core stable, then it also equivalently exhibits the inner stability. Additionally, if the affinity graph weights are symmetric, by Lemma 2 we can directly obtain the result. $\quad\square$

**Theorem 2.** *In a OSB-CAG for ad hoc teamwork, if for all $(j, j') \in \mathcal{E}$, all states $s \in \mathcal{S}$ and all action pairs $(a_j, a_{j'}) \in \mathcal{A}_j \times \mathcal{A}_{j'}$, $w_{jj'}(s, a_j, a_{j'}) \geq 0$, then the grand coalition exhibits strict core stability.*

*Proof.* Since the affinity graph is defined as a star graph with the learner as the internal node, for all states $s \in \mathcal{S}$ and all action pairs $(a_j, a_{j'}) \in \mathcal{A}_j \times \mathcal{A}_{j'}$, under the grand coalition, the learner's preference reward is defined as $R_i(s, a) = \sum_{j \in -i} w_{ij}(s, a_i, a_j)$, while each ad hoc teammate $j$'s preference reward is defined as $R_j(s, a) = w_{ji}(s, a_i, a_j)$.

For the sake of contradiction, for all states $s \in \mathcal{S}$ and all action pairs $(a_j, a_{j'}) \in \mathcal{A}_j \times \mathcal{A}_{j'}$, we assume that when $w_{jj'}(s, a_j, a_{j'}) \geq 0$, there exists some other coalition $\mathcal{C} \subset \mathcal{N}$ leading to that $\exists j_m \in \mathcal{C}, R_j(s, a_c) > R_j(s, a)$. We will discuss problem in the following two cases.

Case 1: If $i \notin \mathcal{C}$, then $R_j(s, a_c) > w_{ji}(s, a_i, a_j)$, which leads to a contradiction to individual rationality.

Case 2: Now, let us discuss the case when $i \in \mathcal{C}$. If the learner $i$ is the $j_m$, then $R_i(s, a_c) = \sum_{j \in \mathcal{C}} w_{ij}(s, a_i, a_j) > R_i(s, a) = \sum_{j \in -i} w_{ij}(s, a_i, a_j)$. The above relation leads to the following relation such that $0 > \sum_{j \in -i \backslash \mathcal{C}} w_{ij}(s, a_i, a_j)$, which contradicts the condition that $w_{jj'}(s, a_j, a_{j'}) \geq 0$. On the other hand, if a teammate $j$ is the $j_m$, then $R_j(s, a_c) = w_{ji}(s, a_j, a_i) > R_j(s, a) = w_{ji}(s, a_j, a_i)$, which is obviously a contradiction.

Following the discussion above, we can conclude that there exists no other coalition $\mathcal{C} \subset \mathcal{N}$ leading to that $\exists j_m \in \mathcal{C}, R_j(s, a_c) > R_j(s, a)$, which results in that there exists no weakly blocking coalition $\mathcal{C}$ as Section 2.1 defines, and therefore the grand coalition exhibits strict core stability. $\qquad \square$

**Proposition 2.** *For the learner $i$ and any teammate $j$ at timestep $t$, suppose that $R_j(s_t, a_{t,j}) = \frac{1}{|-i|} \cdot R_i(s_t, a_{t,i})$ and the affinity graph weights of a star graph are defined as Definition 3 shows, $\alpha_{ij} = \alpha_{ji} \geq 0$, $R_j \geq 0$ and $R_i \geq 0$ would lead to symmetry of affinity weights, individual rationality and the grand coalition exhibiting strict core stability.*

*Proof.* First, we represent affinity graph weights for all edges in the star affinity graph such that $w_{ij}(s_t, a_{t,i}, a_{t,j})$ and $w_{ji}(s_t, a_{t,j}, a_{t,i})$ for all teammates $j$ as follows:

$$w_{ij}(s_t, a_{t,i}, a_{t,j}) = \alpha_{ij}(s_t, a_{t,i}, a_{t,j}) + \frac{1}{2|-i|} \cdot R_i(s_t, a_{t,i}) + \frac{1}{2} \cdot R_j(s_t, a_{t,j}),$$

$$w_{ji}(s_t, a_{t,j}, a_{t,i}) = \alpha_{ji}(s_t, a_{t,j}, a_{t,i}) + \frac{1}{2} \cdot R_j(s_t, a_{t,j}) + \frac{1}{2|-i|} \cdot R_i(s_t, a_{t,i}).$$

It is obvious that if $\alpha_{ij}(s_t, a_{t,i}, a_{t,j}) = \alpha_{ji}(s_t, a_{t,j}, a_{t,i})$, then $w_{ij}(s_t, a_{t,i}, a_{t,j}) = w_{ji}(s_t, a_{t,j}, a_{t,i})$ for all teammates $j \in -i$. As a result, the affinity graph weights would be symmetric.

Next, let us discuss how the representations of $w_{ij}$ and $w_{ji}$ above lead to the individual rationality. Recall that for any agent $j'$, the individual rationality is satisfied if its preference reward $R_{j'}(s_t, a_t) \geq R_{j'}(s_t, a_{t,j'})$ holds. For any teammate $j$, its preference reward for the ad hoc team $R_j(s_t, a_t)$ can be expressed as follows:

$$R_j(s, a) = w_{ji}(s, a_j, a_i) = \alpha_{ji}(s, a_j, a_i) + \frac{1}{2} \cdot R_j(s, a_j) + \frac{1}{2|-i|} \cdot R_i(s, a_i).$$

If $\alpha_{ji}(s_t, a_{t,j}, a_{t,i}) \geq 0$ and $R_j(s_t, a_{t,j}) = \frac{1}{|-i|} \cdot R_i(s_t, a_{t,i})$, then $R_j(s_t, a_t) \geq R_j(s_t, a_{t,j})$.

For the learner $i$, its preference reward for the ad hoc team $R_i(s_t, a_t)$ can be expressed as follows:

$$\begin{aligned}
R_i(s_t, a_t) &= \sum_{j \in -i} w_{ij}(s_t, a_{t,i}, a_{t,j}) \\
&= \sum_{j \in -i} \left\{ \alpha_{ij}(s_t, a_{t,i}, a_{t,j}) + \frac{1}{2|-i|} \cdot R_i(s_t, a_{t,i}) + \frac{1}{2} \cdot R_j(s_t, a_{t,j}) \right\} \\
&= \sum_{j \in -i} \alpha_{ij}(s_t, a_{t,i}, a_{t,j}) + \frac{1}{2} \cdot R_i(s_t, a_{t,i}) + \frac{1}{2} \cdot \sum_{j \in -i} R_j(s_t, a_{t,j}).
\end{aligned}$$

If $\alpha_{ji}(s_t, a_{t,j}, a_{t,i}) \geq 0$ and $R_j(s_t, a_{t,j}) = \frac{1}{|-i|} \cdot R_i(s_t, a_{t,i})$, then $R_i(s_t, a_t) \geq R_i(s_t, a_{t,i})$.

Finally, it is not difficult to observe that $\alpha_{ij}(s_t, a_{t,i}, a_{t,j}) = \alpha_{ji}(s_t, a_{t,j}, a_{t,i}) \geq 0$, $R_j(s_t, a_{t,j}) \geq 0$ and $R_i(s_t, a_{t,i}) \geq 0$ result in that $w_{jj'}(s_t, a_{t,j}, a_{t,j'}) \geq 0$. As a result, it would lead to the grand coalition exhibiting strict core stability as Theorem 2 states. $\qquad \square$

**Lemma 1.** *Suppose that $\alpha_{jj'}(s_t, a_{t,j}, a_{t,j'}) = 0$ for $t \geq T$ if agent $j$ or $j'$ leaves the environment at timestep $T$ and $R_j(s_t, a_{t,j}) = 0$ for $t \geq T$ if agent $j$ leaves the environment at timestep $T$, then $Q_{jj'}(s_t, a_{t,j}, a_{t,j'}) = \mathbb{E}[\sum_{t=0}^{\infty} \gamma^t \alpha_{jj'}(s, a_{t,j}, a_{t,j'})]$ and $Q_j(s_t, a_{t,j}) = \mathbb{E}[\sum_{t=0}^{\infty} \gamma^t R_j(s_t, a_{t,j})]$.*

*Proof.* Suppose that agent $j$ or $j'$ leaves the environment at timestep $T$. Then, we can have the following expression that $Q_{jj'}(s_t, a_{t,j}, a_{t,j'}) = \mathbb{E}[\sum_{t=0}^{\infty} \gamma^t \alpha_{jj'}(s_t, a_{t,j}, a_{t,j'})]$ by the conditions such

that $\alpha_{jj'}(s_t, a_{t,j}, a_{t,j'}) = 0$ if agent $j$ leaves the environment at timestep $t$:

$$
\begin{aligned}
Q_{jj'}(s_t, a_{t,j}, a_{t,j'}) &= \mathbb{E}\Big[\sum_{t=0}^{T} \gamma^t \alpha_{jj'}(s_t, a_{t,j}, a_{t,j'})\Big] \\
&= \mathbb{E}\Big[\sum_{t=0}^{T} \gamma^t \alpha_{jj'}(s_t, a_{t,j}, a_{t,j'}) + \underbrace{\sum_{t=T}^{\infty} \gamma^t \alpha_{jj'}(s_t, a_{t,j}, a_{t,j'})}_{=0}\Big] \\
&= \mathbb{E}\Big[\sum_{t=0}^{\infty} \gamma^t \alpha_{jj'}(s_t, a_{t,j}, a_{t,j'})\Big].
\end{aligned}
$$

Similarly, by the condition that $R_j(s_t, a_{t,j}) = 0$ for $t \geq T$ if agent $j$ leaves the environment at timestep $T$, we can derive the result that $Q_j(s_t, a_{t,j}) = \mathbb{E}[\sum_{t=0}^{\infty} \gamma^t R_j(s_t, a_{t,j})]$. $\qquad \square$

**Theorem 3.** *Suppose that $\alpha_{jj'}(s_t, a_{t,j}, a_{t,j'}) = 0$ for $t \geq T$ if agent $j$ or $j'$ leaves the environment at timestep $T$ and $R_j(s_t, a_{t,j}) = 0$ for $t \geq T$ if agent $j$ leaves the environment at timestep $T$, then the open ad hoc Bellman optimality equation is defined as Eq. 5.*

*Proof.* We derive Eq. 5 as follows. By the results of Lemma 1, we can represent the optimal global Q-value $Q^*(s_t, a_t)$ as follows:

$$
\begin{aligned}
&Q^*(s_t, a_t) \\
&= \mathbb{E}\Big[\sum_{t=0}^{\infty} \gamma^t R(s_t, a_t)\Big|s_t, a_t, \pi^*\Big] = \mathbb{E}\Big[\sum_{t=0}^{\infty} \gamma^t \sum_{j \in \mathcal{N}_t} R_j(s_t, a_t)\Big|s_t, a_t, \pi^*\Big] \\
&= \sum_{j \in \mathcal{N}_t} \mathbb{E}\Big[\sum_{t=0}^{\infty} \gamma^t R_j(s_t, a_t)\Big|s_t, a_t, \pi^*\Big] \\
&= \sum_{j \in \mathcal{N}_t} \mathbb{E}\Big[\sum_{t=0}^{\infty} \gamma^t \Big(\sum_{j' \in N_t(j)} \alpha_{jj'}(s, a_{t,j}, a_{t,j'}) + R_j(s_t, a_{t,j})\Big)\Big|s_t, a_t, \pi^*\Big] \qquad (9) \\
&= \sum_{j \in \mathcal{N}_t} \Big\{\sum_{j' \in N_t(j)} \mathbb{E}\Big[\sum_{t=0}^{\infty} \gamma^t \alpha_{jj'}(s, a_{t,j}, a_{t,j'})\Big|s_t, a_t, \pi^*\Big] + \mathbb{E}\Big[\sum_{t=0}^{\infty} \gamma^t R_j(s_t, a_{t,j})\Big|s_t, a_t, \pi^*\Big]\Big\} \\
&= \sum_{j \in \mathcal{N}_t} \Big\{\sum_{j' \in N_t(j)} Q_{jj'}^*(s_t, a_{t,j}, a_{t,j'}) + Q_j^*(s_t, a_{t,j})\Big\} = \sum_{j \in \mathcal{N}_t} Q_j^*(s_t, a_t),
\end{aligned}
$$

where $a_t \in \times_{j \in \mathcal{N}} \mathcal{A}_j$, $\pi^* = \times_{j \in \mathcal{N}_t} \pi_j^*$ and $N_t(j)$ indicates the neighbours of agent $j$ at timestep $t$. Due to the convexity of the max operator, we can get the following relation such that

$$
\begin{aligned}
\max_{a_i'} \mathbb{E}[Q^*(s_{t+1}, a_{t+1,-i}, a_i')] &= \max_{a_i'} \mathbb{E}\Big[\sum_{j \in \mathcal{N}_{t+1}} Q_j^*(s_{t+1}, a_{t+1,-i}, a_i')\Big] \\
&\leq \sum_{j \in \mathcal{N}_{t+1}} \max_{a_i'} \mathbb{E}\big[Q_j^*(s_{t+1}, a_{t+1,-i}, a_i')\big].
\end{aligned} \qquad (10)
$$

There exists $b(s_{t+1}) \geq 0$ for all $s_{t+1} \in \mathcal{S}$ enabling the above result to be equivalently written as that

$$
\max_{a_i'} \mathbb{E}[Q^*(s_{t+1}, a_{t+1,-i}, a_i')] + b(s_{t+1}) = \sum_{j \in \mathcal{N}_{t+1}} \max_{a_i'} \mathbb{E}\big[Q_j^*(s_{t+1}, a_{t+1,-i}, a_i')\big]. \qquad (11)
$$

According to Assumption 2 that *the learner is able to influence the ad hoc teammates (expressed as preference rewards)*, we can expand the optimal preference Q-value of each agent $j \in \mathcal{N}_t$ as per the fashion of the Bellman optimality equation such that

$$
Q_j^*(s_t, a_t) = R_j(s_t, a_t) + \gamma \mathbb{E}\Big[\max_{a_i'} \mathbb{E}\big[Q_j^*(s_{t+1}, a_{t+1,-i}, a_i')\big]\Big], \qquad (12)
$$

where $Q_j^*(s_{t+1}, a_{t+1,-i}, a_i') = 0$ if agent $j$ leaves at timestep $t+1$ by Lemma 1.

Based on Eq. 12, we can sum up all possible agents and get a condition necessary for achieving the optimality of all surviving agents belonging to $\mathcal{N}_t$ such that

$$
\sum_{j \in \mathcal{N}_t} Q_j^*(s_t, a_t) = \sum_{j \in \mathcal{N}_t} R_j(s_t, a_t) + \gamma \mathbb{E}\Big[\sum_{j \in \mathcal{N}_{t+1}} \max_{a_i'} \mathbb{E}\big[Q_j^*(s_{t+1}, a_{t+1,-i}, a_i')\big]\Big]. \qquad (13)
$$

Note that Eq. 13 does not hold if $\mathcal{N}_t \subset \mathcal{N}_{t+1}$, since it is unreasonable to consider an agent $j \in \mathcal{N}_{t+1}$ but $\notin \mathcal{N}_t$ at timestep $t$, which can be seen as a singularity of this Bellman optimality equation. In other words, Eq. 13 is undefined over $P_T(\mathcal{N}_{t+1}, s_{t+1}|\mathcal{N}_t, s_t, a_t)$ when $\mathcal{N}_t \subset \mathcal{N}_{t+1}$.

Substituting Eq. 9 and Eq. 11 into Eq. 13, by the relation that $R(s_t, a_t) = \sum_{j \in \mathcal{N}_t} R_j(s_t, a_t)$ we can obtain Eq. 5. $\qquad \square$

**Remark 2.** *The open team setting in Lemma 1 and Theorem 3 can be understood as that the timesteps at which all relevant agents join the environment are scheduled at the beginning. Thereby, we can regard the composition of an agent label and its timestep to join the environment as an index to differentiate all possible such compositions. Note that if an agent joins the environment twice, it would be two different indices to describe these two events. The length of an episode can be assumed to a very large number that is almost infinite but still finite such as $10^{12}$, so that the number of compositions must be finite. All compositions are set to the inactive status. If a composition is activated at some timestep, then it would receive its normal preference reward until it leaves the environment and becomes inactive again. Otherwise, its preference reward would be zero. Owing to that the switch of preference rewards can be seen as part of the environmental operations, the common Bellman optimality equation and Bellman operator can be used to describe and solve this problem. About the relevent Bellman optimality equation, we can well define a finite number of optimal preference Q-values indexed by the compositions defined above at any timestep and state. The optimal global Q-value at each timestep only takes the active compositions into consideration without loss of generality to describe the optimality of all compositions as a necessary condition. One can check that the relevant Bellman operator (dynamic programming) can be defined as the similar manner to reach the Bellman optimality equation defined above.*

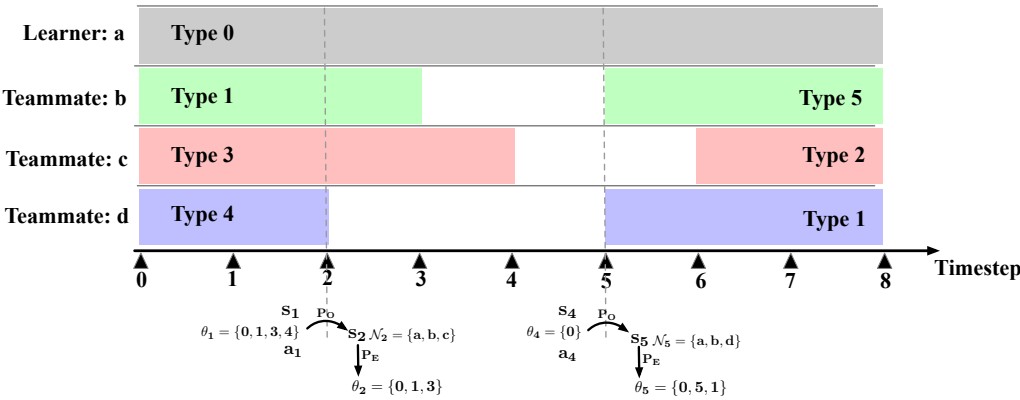

Figure 4: Illustration of open ad hoc teamwork with up to 3 teammates existing in the environment. Each row represents the placeholder for the corresponding roles, where the white interval indicates the period when the teammate is absent. The colored interval represents the survival of a teammate played by an agent with an assigned agent type (noted by the text "Type #") sampled from the agent set formed at the beginning of the whole process. Below the horizontal axis, we present two examples to justify $P_O$ and $P_E$ that form the transition function $T$. The example on the left shows the transition of the team from timestep 1 to timestep 2. At timestep 1, the agent-type set $\theta_1$ is $\{0, 1, 3, 4\}$, which is assigned to learner $a$ and three teammates denoted by $b$, $c$, and $d$ respectively. As per Eq. 2, $s_2, \mathcal{N}_2 = P_O(s_1, a_1, \theta_1)$, where $\mathcal{N}_2$ includes the members $a, b, c$. $P_E$ then assigns the agent types $\theta_2 = \{0, 1, 3\}$ in terms of $s_2, \mathcal{N}_2$, such that $\theta_2 = P_E(s_2, \mathcal{N}_2)$. Similarly, the example on the right illustrates the team's expansion with the addition of two new members. At timestep 4, the team only consists of the learner. Then, adding three teammates $a$, $b$ and $d$ at timestep 5.

**Proposition 3.** *$P_E(\theta_t|\mathcal{N}_t, s_t)$ can be approximated as the expression $\hat{P}_E(\theta_t|s_t, \theta^-_{t-1}, \theta^+_{t-1})$, where $\theta^-_{t-1}$ indicates the types of remaining agents of $\mathcal{N}_{t-1}$ and $\theta^+_{t-1}$ indicates the unknown types of newly joined agents at timestep $t$ (set to zeros as default).*

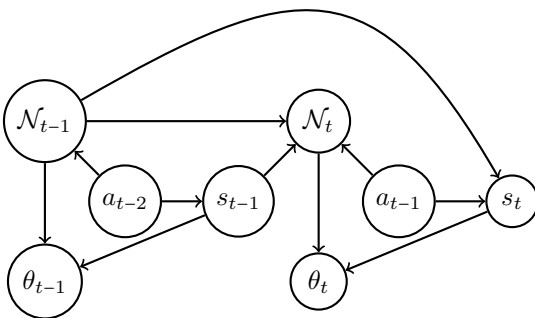

Figure 5: Causal diagram encoding the conditional independence existing in the problem.

*Proof.* At the beginning, we construct a causal diagram in Fig.5 that represents all probability distributions establishing the OSBHG such that

$$P_E(\theta_t|\mathcal{N}_t, s_t),$$
$$P_T(\mathcal{N}_t, s_t|\mathcal{N}_{t-1}, s_{t-1}, a_{t-1}),$$
$$P(\mathcal{N}_{t-1}|\theta_{t-1}, s_{t-1}).$$

The above probability distributions are the most compact representation which cannot be simplified by the d-Separation criterion Pearl (2009). We can say that *the causal diagram is an instance to preserve the information in our problem* and therefore it is faithful to use this causal diagram for approximating $P_E(\theta_t|\mathcal{N}_t, s_t)$.

Now, we begin the approximation. First, let the set of agents at timestep $t$ be factorized to two parts $\Delta\mathcal{N}_t$ and $\mathcal{N}_{t-1}$ such that $\mathcal{N}_t = \Delta\mathcal{N}_t^+ \cup (\mathcal{N}_{t-1} - \Delta\mathcal{N}_t^-)$, where $\Delta\mathcal{N}_t^+ = \mathcal{N}_t - \mathcal{N}_{t-1}$ indicates the new agents appears at timestep $t$; $\Delta\mathcal{N}_t^- = \mathcal{N}_{t-1} - \mathcal{N}_t$ indicates the agents exist at timestep $t-1$ but leave at timestep $t$ and $\mathcal{N}_{t-1} - \Delta\mathcal{N}_t^-$ indicates the remaining agents of $\mathcal{N}_{t-1}$ at timestep $t$. From the causal diagram, it is apparent that $\theta_{t-1}$ is a descendent variable (child) of $\mathcal{N}_{t-1}$ with close correlation. Referring to the common proxy variable selection strategy in the causal inference community (Wang & Blei, 2021) that *an observed child can be used as a proxy of an unobserved variable*, $\theta_{t-1}$ can be used as a proxy of $\mathcal{N}_{t-1}$ to reflect the information that is sufficient to represent agents, i.e. agent types. Using $\theta_{t-1}$, it is easy to express the information of $\mathcal{N}_{t-1} - \Delta\mathcal{N}_t^-$ by removing the $\theta_{t-1,k}, \forall k \in \Delta\mathcal{N}_t^-$ from $\theta_{t-1}$, denoted as $\theta_{t-1}^-$. Since there is no observation or evidence to support and reflect $\Delta\mathcal{N}_t^+$, it is expressed as a default value, e.g., each agent belonging to $\Delta\mathcal{N}_t^+$ is equipped with a zero vector, denoted as $\theta_{t-1}^+$. Replacing $\mathcal{N}_t$ by its proxy $(\theta_{t-1}^-, \theta_{t-1}^+)$, we obtain a new probability distribution $\hat{P}_E(\theta_t|s_t, \theta_{t-1}^-, \theta_{t-1}^+)$. $\qquad\square$

## E  ADDITIONAL THEORETICAL RESULTS

**Proposition 4.** *For any agent $j$ at timestep $t$, if the affinity graph is a fully connected graph and agents' individual preference rewards are identical, there exists a representation of $w_{jj'}$, for which $\alpha_{jj'} = \alpha_{j'j} \geq 0$ and $R_j \geq 0$ would lead to symmetry of affinity weights, individual rationality and the grand coalition exhibiting strict core stability.*

*Proof.* First, we represent affinity graph weights in the fully connected affinity graph such as $w_{jj'}(s_t, a_{t,j}, a_{t,j'})$ for all agents $j$ and its neighbours $j'$ as follows:

$$w_{jj'}(s_t, a_{t,j}, a_{t,j'}) = \alpha_{jj'}(s_t, a_{t,j}, a_{t,j'}) + \frac{1}{2|-j|} \cdot R_j(s_t, a_{t,j}) + \frac{1}{2|-j|} \cdot R_{j'}(s_t, a_{t,j'}),$$

where $-j$ denotes the agent set excluding $j$ at timestep $t$. It is not difficult to observe that if $\alpha_{jj'}(s_t, a_{t,j}, a_{t,j'}) = \alpha_{j'j}(s_t, a_{t,j'}, a_{t,j})$, then $w_{jj'}(s_t, a_{t,j}, a_{t,j'}) = w_{j'j}(s_t, a_{t,j'}, a_{t,i})$ for all agents $j \in -i$ and its neighbours $j'$. As a result, the affinity graph weights would be symmetric.

Next, let us discuss how the representations of $w_{jj'}$ and $w_{j'j}$ above lead to the individual rationality. Recall that for any agent $j$, the individual rationality is satisfied if its preference reward $R_j(s_t, a_t) \geq$

$R_j(s_t, a_{t,j})$ holds. For any agent $j$, its preference reward for the ad hoc team $R_j(s_t, a_t)$ can be expressed as follows:

$$R_j(s_t, a_t) = \sum_{j' \in -j} w_{jj'}(s_t, a_{t,j}, a_{t,j'})$$

$$= \sum_{j' \in -j} \alpha_{jj'}(s_t, a_{t,j}, a_{t,j'}) + \frac{1}{2} \cdot R_j(s_t, a_{t,j}) + \frac{1}{2|-j|} \cdot \sum_{j' \in -j} R_{j'}(s_t, a_{t,j'}).$$

If $\alpha_{ji}(s_t, a_{t,j}, a_{t,i}) \geq 0$ and $R_j(s_t, a_{t,j}) = R_{j'}(s_t, a_{t,j'})$, then $R_j(s_t, a_t) \geq R_j(s_t, a_{t,j})$.

Finally, it is not difficult to observe that $\alpha_{jj'}(s_t, a_{t,j}, a_{t,j'}) = \alpha_{j'j}(s_t, a_{t,j'}, a_{t,j}) \geq 0$ for any agent $j$ and its neighbour $j'$ and $R_j(s_t, a_{t,j}) \geq 0$ for any agent $j$ result in that $w_{jj'}(s_t, a_{t,j}, a_{t,j'}) \geq 0$. As a result, it would lead to the grand coalition exhibiting strict core stability as Theorem 2 states.  □

## F  THE GPL FRAMEWORK

We now review the GPL framework (Rahman et al., 2021). GPL consists of the following modules: type inference model, joint action value model and agent model. We only summarize the model specifications. About other details, please refer to the original paper of GPL.

**Type Inference Model.** This is modelled as a LSTM (Hochreiter & Schmidhuber, 1997) to infer agent types of a team at timestep $t$ given that of a team at timestep $t-1$. The agent type is modelled as a fixed-length hidden-state vector of LSTM, named as agent type embedding. As each timestep $t$, the state information of an emergent team $\mathcal{N}_t$ is reproduced to a batch of agents' information $B_t = [\langle u_t, x_{t,1}\rangle, ..., \langle u_t, x_{t,|\mathcal{N}_t|}\rangle]^\top$, where each agent is preserved a vector composing $u_t$ and $x_{t,i}$ which are observations and agent specific information extracted from state $s_t$. Along with additional information such as the agent type embedding of $\mathcal{N}_{t-1}$ and the cell state, LSTM estimates the agent type embedding of $\mathcal{N}_t$. To address the situation of changing team size, at each timestep the type embedding of the agents who leave a team would be removed, while the new added agents' type embedding would be set to a zero vector.

**Joint Action Value Model.** The joint action value $Q_{\beta,\delta}(s_t, a_t)$ is approximated as the sum of individual utility terms $Q_\beta^j(a_{t,j}|s_t)$ and pairwise utility terms $Q_\delta^{j,k}(a_{t,j}, a_{t,k}|s_t)$ such that

$$Q_{\beta,\delta}(s_t, a_t) = \sum_{j \in \mathcal{N}_t} Q_\beta^j(a_{t,j}|s_t) + \sum_{\substack{j,k \in \mathcal{N}_t \\ j \neq k}} Q_\delta^{j,k}(a_{t,j}, a_{t,k}|s_t). \tag{14}$$

Both $Q_\beta^j(a_{t,j}|s_t)$ and $Q_\delta^{j,k}(a_{t,j}, a_{t,k}|s_t)$ are implemented as multilayer perceptrons (MLPs) parameterised by $\beta$ and $\delta$, denoted as $\text{MLP}_\beta$ and $\text{MLP}_\delta$. The input of $\text{MLP}_\beta$ is the concatenation of the ad hoc agent's type embedding $\theta_{t,i}$ and the target agent's type embedding $\theta_{t,j}$, and its output is a vector with the length of $|\mathcal{A}_j|$ estimating $Q_\delta^{j,k}(a_{t,j}, a_{t,k}|s_t)$ such that

$$Q_\beta^j(a_{t,j}|s_t) = \text{MLP}_\beta(\theta_{t,j}, \theta_{t,i})(a_{t,j}). \tag{15}$$

$Q_\delta^{j,k}(a_{t,j}, a_{t,k}|s_t)$ is approximated by low-rank factorization such that

$$Q_\delta^{j,k}(a_{t,j}, a_{t,k}|s_t) = (\text{MLP}_\delta(\theta_{t,j}, \theta_{t,i})^\top \text{MLP}_\delta(\theta_{t,k}, \theta_{t,i}))(a_{t,j}, a_{t,k}), \tag{16}$$

where the input of $\text{MLP}_\delta$ is the same as $\text{MLP}_\beta$; the output of $\text{MLP}_\delta(\theta_{t,j}, \theta_{t,i})$ is a matrix with the shape $K \times |\mathcal{A}_j|$ and $K \ll |\mathcal{A}_j|$.

**Agent Model.** It is assumed that all other agents would affect an agent's actions. To model this situation, GNNs are applied to process the agent type embedding of an ad hoc team $\theta_t$, where each agent is seen as a node. More specifically, a GNN model called relational forward model (RFM) (Tacchetti et al., 2019) parameterised by $\eta$ is applied to transform $\theta_t$ as initial node representation to $\bar{n}_t$ as new node representation considering other agents' effects. Then, $\bar{n}_t$ is used to infer $q_{\zeta,\eta}(a_{t,-i}|s_t)$ as the approximation of teammates' joint policy $\pi_{t,-i}(a_{t,-i}|s_t, \theta_{t,-i})$ such that

$$q_{\zeta,\eta}(a_{t,-i}|s_t) = \prod_{j \in -i} q_{\zeta,\eta}(a_{t,j}|s_t),$$

$$q_{\zeta,\eta}(a_{t,j}|s_t) = \text{Softmax}(\text{MLP}_\eta(\bar{n}_{t,j}))(a_{t,j}). \tag{17}$$

**Learner's Action Value.** Substituting the agent model and the joint action value model defined above into Eq. 1, the ad hoc agent's action value can be approximated as follows:

$$Q^{\pi_i}(s_t, a_{t,i}) = Q^i_\beta(a_{t,j}|s_t) + \sum_{a_j \in \mathcal{A}_j, j \neq i} (Q^j_\beta(a_{t,j}|s_t) + Q^{i,j}_\beta(a_{t,i}, a_{t,j}|s_t)) q_{\zeta,\eta}(a_{t,j}|s_t)$$
$$+ \sum_{\substack{a_{t,j} \in \mathcal{A}_j, a_{t,k} \in \mathcal{A}_k, \\ j,k \neq i}} Q^{j,k}_\delta(a_{t,j}, a_{t,k}|s_t) q_{\zeta,\eta}(a_{t,j}|s_t) q_{\zeta,\eta}(a_{t,j}|s_t). \quad (18)$$

## G  EXPERIMENTAL SETUPS

We evaluate the proposed CA-GPL in two existing environments LBF and Wolfpack with open team settings (Rahman et al., 2021; 2022), where teammates are randomly selected to enter the environment and stay for a certain number of timesteps. If a teammate has stayed for longer timesteps than its allocated lifetime, it would be removed from the environment and re-allocated to a re-entry queue with a random waiting time. The randomized re-entry queue directly leads to different compositions of teammates in an ad hoc team. If the number of agents existing in the environment does not reach the upper limit, the agents in the re-entry queue would be pushed into the environment. Specifically, within the Wolfpack environment, we uniformly determine the active duration, selecting a value between 25 and 35 timesteps, while the dead duration is uniformly sampled between 15 and 25 timesteps. Conversely, the durations for LBF are somewhat shorter, with the active duration uniformly sampled between 15 and 25 timesteps, and the dead duration between 10 and 20 timesteps.

The teammate policies follow the setting of GPL Rahman et al. (2021), which implement a set of heuristic policies as teammates. Specifically, in Wolfpack environment, the teammate set consists of Random agent, Greedy agent, Greedy probabilistic agent, Teammate aware agents, GNN-Based teammate aware agents, Graph DQN agents, Greedy waiting agents, Greedy probabilistic waiting agents, Greedy team-ware waiting agents. With LBF, a mixture of heuristics and A2C agent are used as the teammate policy set. Detailed information about teammate set could be found in Appendix B.4 of Rahman et al. (2021). Since the provided A2C agent from GPL only supports the scenarios with up to 4 teammates, we additionally train an A2C agent for the scenarios with up to 8 teammates.

The detailed common training parameters of CA-GPL related algorithms are provided as follows.

- `lr:  0.00025`
- `gamma:  0.99`
- `max_num_steps:  400000`
- `eps_length:  200 for LBF / 2000 for Wolfpack`
- `update_frequency:  4`
- `saving_frequency:  50`
- `num_envs:  16`
- `tau:  0.001`
- `eval_eps:  5`
- `weight_predict:  1.0`
- `num_players_train:  3`
- `num_players_test:  5 for scenarios with up to 4 teammates / 9 for scenarios with up to 8 teammates`
- `seed:  0`
- `eval_init_seed:  2500`
- `close_penalty:  0.5 for Wolfpack / None for LBF`

Then, we report the exclusive parameters of CA-GPL and its variants as follows.

- **CA-GPL:** `star_graph=True, pos_pair=True, pos_indiv=True`
- **CA-GPL-NI:** `star_graph=True, pos_pair=True, neg_indiv=True`

- CA-GPL-NP: `star_graph=True, neg_pair=True, pos_indiv=True`
- CA-GPL-NPQ: `star_graph=True, neg_pair=True, neg_indiv=True`
- CA-GPL-ZI: `star_graph=True, pos_pair=True, zero_indiv=True`
- CA-GPL-FCG: `pos_pair=True, neg_indiv=True`

The parameters not mentioned for a method above are set to `False` as default.

All experiments are run on Xeon Gold 6230 with 20 CPU cores. An experiment on Wolfpack takes around 8 hours, while an experiment on LBF takes around 5 hours.

## H    ADDITIONAL EXPERIMENTAL RESULTS

### H.1    ADDITIONAL EVALUATION ON GRAPH STRUCTURES

Fig. 6 compares the distinct graph structures for GNNs - the star graph and the fully connected graph, used to aggregate observations for teammates' policies in both LBF and Wolfpack, with settings allowing for up to 8 teammates. As depicted in the figure, the CA-GPL employing a star graph surpasses the fully connected graph CA-GPL (CA-GPL-FCPG). This not only verifies but also reinforces our approach of modeling the affinity graph (coordination graph) as a star graph.

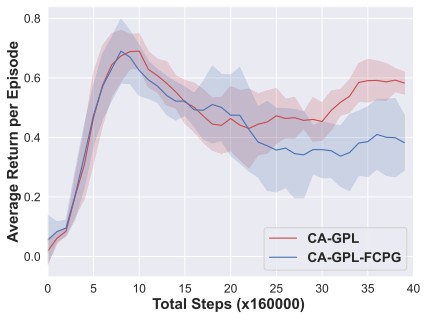
(a) LBF up to 8 teammates.

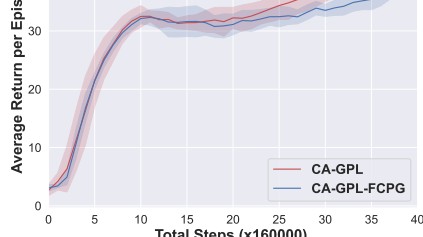
(b) Wolfpack up to 8 teammates.

Figure 6: Additional comparison between different graph structures for GNNs to aggregate observations for teammates' policies in LBF and Wolfpack environments with up to 8 teammates.

### H.2    CASE STUDY: COOPERATIVE BEHAVIORS AND PREFERENCE Q-VALUE ANALYSIS

Fig. 7 depicts the trajectories and associated agents' preference Q-values (shortened as Pref-Q). The trajectories are sampled from the LBF setting with up to 2 teammates. In the depicted keyframes, arrows indicate the current movement. The numbers on the apple icon and arrows denote the level; players can collect (marked with a star symbol) the apple if the sum of the players' levels is no less than the level of the apple. Besides, small circles represent stationary actions. Below the time axis, the heat maps display Pref-Q, with the x-axis representing the actions. The last row of each heatmap indicates the learner's preference Q-value, while the previous rows depict teammates' Pref-Q, given teammates' actual actions. The 'tri_up' marker signifies the current action of the learner. As illustrated in Fig. 7, the learner successfully cooperates with variable teammates. Initially, two teammates are randomly generated; however, Teammate 1 is removed as seen in Frame ii. In the final phase, the learner strategically remains stationary, allowing collaborative apple collection with Teammate 2, as depicted in Frames iii and iv. Furthermore, the heatmaps showcase the alignment between Pref-Q and cooperative behaviors, even within a variable team setting. Nevertheless, the preference Q-value cannot always perfectly align with the learner's actions due to the learner's inaccurate estimation of teammates' policies. The most successful alignments between trajectories and Pref-Q justify the efficacy of Pref-Q to encourage collaboration in open ad hoc teamwork and, thus, verify the reasonableness of our theoretical model. Additional demos supporting these conclusions and high-resolution Fig. 7 can be found on our homepage, shown in Abstract.

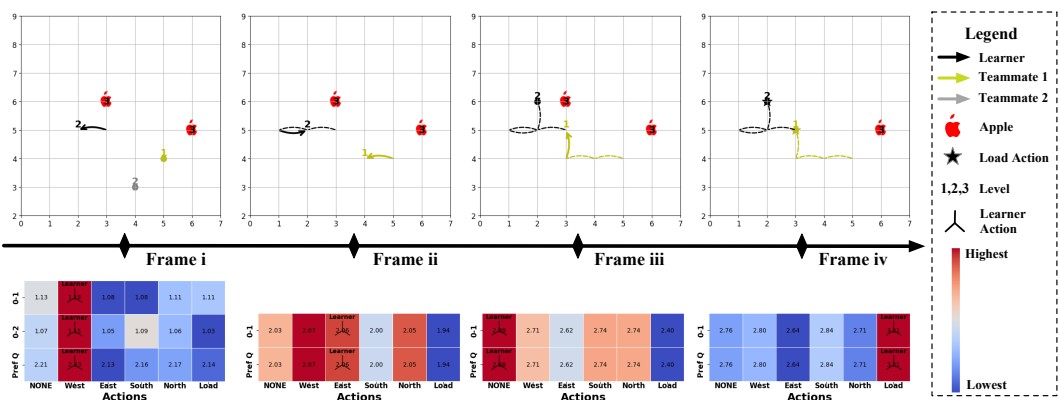

Figure 7: Visualization of cooperative behaviors and preference Q-values on LBF. The four frames above the time-axis correspond to the four sets of preference Q-values shown below the axis. In the depicted keyframes of trajectory visualization, arrows indicate the current movement. The numbers on the apple icon and arrows denote the level; players can collect (marked with a star symbol) the apple if the sum of the players' levels is no less than the level of the apple. Besides, small circles represent stationary actions. In heat maps, the redder the color, the greater the Q value. It is observable that the learner's actions, marked by "tri_up", are consistently aligned with the areas of highest Q values. The most successful alignments between trajectories and Pref-Q justify the efficacy of Pref-Q to encourage collaboration in open ad hoc teamwork and, thus, verify the reasonableness of our theoretical model.

