# OpenReview forum: "A Cooperative-Game-Theoretical Model for Ad Hoc Teamwork"
_ICLR.cc/2024/Conference — Submitted to ICLR 2024_

### Official Review · Reviewer_eRVD · 2023-10-15

**Soundness:** 2 fair
**Presentation:** 1 poor
**Contribution:** 2 fair
**Rating:** 3
**Confidence:** 2

**Summary:**

The authors propose a framework for multi-agent ad-hoc teamwork based around cooperative game theory. They define solution concepts and algorithms based on this concept, and run extensive experiments to compare their method to previous methods.

**Strengths:**

I like the idea of using cooperative game theory to analyze ad-hoc cooperation. The ideas in the paper seem, to the best of my understanding, novel. The experimental results also seem fairly strong, but my ability to understand their significance is limited (see next section).

**Weaknesses:**

I am not too familiar with cooperative game theory, and I found the technical exposition rather hard to follow, and things took a long time for me to parse---perhaps because, while reading, I didn't have a mental model for where things were going/what to expect. I think the exposition would be greatly strengthed by the addition of a running example that the authors could use to demonstrate the various claims and definitions that they make, and by formalizing various definitions mathematically. An incomplete list of specific clarity concerns is listed in the "Questions" section.

I gave up on attempting to parse the rest of the technical part of the paper as I have already spent considerable time and am still confused about several things that are quite fundamental (again, see "Questions" below for some of these). I think there could be an interesting contribution here, but the quality of writing needs to be improved before publication.

From what I can tell, the experimental results seem strong, but my ability to understand the significance of the results is essentially limited to seeing that CA-GPL's line is higher than GPL's.


Nitpicks/minor errors (not affecting score):
* I think $\mathcal A_{\mathcal N}$ and $\Theta_{\mathcal N}$ should have an $\exists$ quantifier in their definitions, not a $\forall$---otherwise, they're both empty sets, since there is not a joint action that is simultaneously in $\mathcal A_{\mathcal N_t}$ for all ${\mathcal N_t}$.
* In the formulation in Sec 2.2, it is unclear how the team $\mathcal N_t$ evolves with time. It seems to me that this is formalized in the next subsection, but if so there should be a forward pointer.

**Questions:**

1. In Theorem 1 it should be made clear exactly what "maximizing the social welfare under the grand coalition" means. I am interpreting it as finding a *joint* policy, that is, a map $\pi : \mathcal S \times \mathbb P(\mathcal N) \ni (s_t, \mathcal N_t) \mapsto a_t \in \mathcal A_{\mathcal N_t}$ that, for every pair $(s_t, \mathcal N_t)$, selects $a_t$ to maximize the local social welfare $\sum_{i \in \mathcal N_t} R_i(s_t, a_t)$. Is that correct? In any case this should be formally stated.
2. It seems like the paper is adopting a single-agent perspective, where there is a single learner $i \in \mathcal N$ and all other agents' policies are held fixed (is this correct?). But then how can we optimize social welfare, required by Theorem 1, if we only control the single learner $i$? In particular, what if the other agents are acting in such a way that learner $i$ alone cannot achieve social welfare optimality?
1. At the beginning of Sec 3.3, I don't understand the point of defining $\mathcal{CS}_t$ only to later set $\mathcal{CS}_t := \mathcal N_t$. Does this have a purpose? It seems cleaner to just not define the extra symbol.
1. In section "Representation of Preference Q-Values", there is a clause beginning "which is presumed ... goal". What does "which" refer to here? Is this an assumption required by Theorem 2? If so it should be formalized mathematically.
1. What's the purpose of the types? They don't seem to be doing anything, except perhaps affecting transitions---but my interpretation of Theorem 1 implies that we only need to perform local optimizations at each state anyway. Perhaps it would be cleaner---and just as interesting---to write the paper without types?

---

> ### Author Response · Authors · 2023-11-17
> **Response to Reviewer eRVD (1/2)**
>
> 1. **In Theorem 1 it should be made clear exactly what "maximizing the social welfare under the grand coalition" means. I am interpreting it as finding a joint policy, that is, a map $\pi: \mathcal{S} \times \mathbb{P}(\mathcal{N}) \in (s\_{t}, \mathcal{N}\_{t}) \mapsto a\_{t} \in \mathcal{A}\_{\mathcal{N}\_{t}}$ that, for every pair $(s\_{t}, \mathcal{N}\_{t})$, selects $a\_{t}$ to maximize the local social welfare $\sum\_{j \in \mathcal{N}\_{t}} R_{j}(s\_{t}, a\_{t})$. Is that correct? In any case this should be formally stated.**
>
>     *Reply:* Thanks for your suggestion. Your understanding about the result of Theorem 1 for fitting the problem we would like to solve (open ad hoc teamwork) is correct. However, the result from Theorem 1 is general for any preference value profile in CAG. Note that in the original version of CAG, it does not consider action space, while in our theory we extend the definition of preference value with action as an decision variable (i.e. a realization of preference value). Nevertheless, we agree with you that the statement should be linked to our problem more directly to make it better comprehensive. For this reason, we have added a corollary to comprehend the result of Theorem 1 in our case in the revised paper.
>
>
> 2. **It seems like the paper is adopting a single-agent perspective, where there is a single learner $i \in \mathcal{N}$ and all other agents' policies are held fixed (is this correct?). But then how can we optimize social welfare, required by Theorem 1, if we only control the single learner $i$? In particular, what if the other agents are acting in such a way that learner $i$ alone cannot achieve social welfare optimality?**
>
>     *Reply:* Thanks for your really thoughtful questions. Yes, in the common setting of ad hoc teamwork, we can only control one agent to collaborate with other teammates on the fly. To make the theoretical analysis simpler, we here assume all other agents' policies are held fixed (i.e., to keep the environment stationary to the learner we control). However, this work can be extended to scenarios where other agents' policies would vary, for which the analysis would become complicated due to the non-stationary envrionment.
>
>     Although we only control the learner $i$, it can affect other agents' decision, since other agents are able to respond the learner's action with regarding the learner's behavior as partial input of their policies. As a result, other agents' individual preference reward as an evaluation of their policies would accordingly change.
>
>     The final question is really insightful. Since the basic assumption of ad hoc teamwork is that all agents are assumed to have a common goal, so the situation you consider is out of scope. Nevertheless, if the situation you state really happens (that is an open question), then for the moment we can only say it could be extended to control a group of agents, where at least one agent is able to influence other uncontrollable agents. We hope our response addresses your concerns.
>
>
> 3. **At the beginning of Sec 3.3, I don't understand the point of defining $\mathcal{CS}\_{t}$ only to later set $\mathcal{CS}\_{t} = \mathcal{N}\_{t}$. Does this have a purpose? It seems cleaner to just not define the extra symbol.**
>
>     *Reply:* Thanks for your suggestion. We agree that your suggestion would make the definition more concise. However, the definition of $\mathcal{CS}_{t}$ here has two purposes. The main purpose is that in the future work this theoretical framework can be directly used to extend to the scenarios where multiple agents divided into different coalitions (e.g., to solve multi-task problems) can be controlled. The second purpose is that we would like to keep the tight connection to the past works. This would make the readers understand the overall relationship between our work and past works, which avoids the potential of reinvention in the future.
>
> 4. **In section "Representation of Preference Q-Values", there is a clause beginning "which is presumed ... goal". What does "which" refer to here? Is this an assumption required by Theorem 2? If so it should be formalized mathematically.**
>
>     *Reply:* Sorry for the confusion. This "which" refers to the condition in Theorem 2 such that $w_{jj'} \geq 0$. We have removed it in the revised paper to keep conciseness and avoid the confusion, since even if $w_{jj'} \geq 0$ does not hold, we can still manually add bias during learning without changing the final result as we discussed in Section 4.4 in the revised paper about practical implementation. Thank you for your mention.

---

> > ### Author Response · Authors · 2023-11-17
> > **Response to Reviewer eRVD (2/2)**
> >
> > 5. **What's the purpose of the types? They don't seem to be doing anything, except perhaps affecting transitions---but my interpretation of Theorem 1 implies that we only need to perform local optimizations at each state anyway. Perhaps it would be cleaner---and just as interesting---to write the paper without types?**
> >
> >     *Reply:* Thank you for the thoughtful comment. Type is the convention in Bayesian games used to describe the **uncertain agents**, which was also used in the original GPL paper. For instance, in the context of ad hoc teamwork, the learner need to infer the behaviors of uncertain teammates. If removing type, it becomes difficult to define the process of inferring uncertain agents' behaviors. On the other hand, rather than use just agent set to represent agents, the definition of types can further help define the varying types of agents (rather than the fixed types considered in this paper) for the future work.
> >
> > 6. **I think $\mathcal{A}\_{\mathcal{N}}$ and $\Theta\_{\mathcal{N}}$ should have an $\exists$ quantifier in their definitions, not a $\forall$ --- otherwise, they're both empty sets, since there is not a joint action that is simultaneously in $\mathcal{A}\_{\mathcal{N}\_{t}}$ for all $\mathcal{N}\_{t}$.**
> >
> >     *Reply:* Thank you for your suggestion. The original expression of $\mathcal{A}\_{\mathcal{N}}$ and $\Theta\_{\mathcal{N}}$ are actually confused. In our view, it could be better to express them as $\mathcal{A}\_{\scriptscriptstyle\mathcal{N}} = \bigcup\_{\mathcal{N}\_{t} \in \mathbb{P}(\mathcal{N})} \{a \vert a \in \mathcal{A}\_{{\scriptscriptstyle \mathcal{N}}\_{t}} \}$ and $\Theta\_{\scriptscriptstyle\mathcal{N}} = \bigcup\_{\mathcal{N}\_{t} \in \mathbb{P}(\mathcal{N})} \{\theta \vert \theta \in \Theta^{{\scriptscriptstyle |\mathcal{N}\_{t}|}} \}$. We are happy to hear from your feedback.
> >
> > 7. **In the formulation in Sec 2.2, it is unclear how the team $\mathcal{N}_{t}$ evolves with time. It seems to me that this is formalized in the next subsection, but if so there should be a forward pointer.**
> >
> >     *Reply:* Thank you for your suggestion. However, $\mathcal{N}\_{t}$ in Section 2.2 was defined by the GPL paper. We agree with you that it is unclear how $\mathcal{N}\_{t}$ evolves. This is the reason why we redefine the dynamics thouroughly in our own theoretical model in the next subsection. We believe you have already discovered some benefits and novelties of our work. To respect the GPL paper and distinguish our work from theirs, we decide to keep the current expression.

---

> > > ### Comment · Reviewer_eRVD · 2023-11-17
> > >
> > > My opinion of the paper has not changed, and I will keep my score.
> > >
> > > There seems to be a strong dependency on the GPL paper in this paper. As a general piece of advice, papers should make effort in writing where possible to reduce such dependency and be self-contained; I would advise the authors to do this, especially for readers (like myself) unfamiliar with GPL.
> > >
> > > **2)** (Single-agent perspective) This remark from the authors raises more questions to me than it answers. The assumption that all other agents are playing a fixed policy completely defeats the purpose of investigating a multi-agent system in the first place---you can treat agents with fixed policies as part of the environment and just solve the resulting (single-agent) POMDP.  If this paper is really about a single-agent perspective, the authors should do much more to justify why we should believe that this single-agent perspective is relevant to a multi-agent system. (For example, if all agents run your algorithm, what happens? There is a lot of work investigating multi-agent learning dynamics in games, and I believe that the authors should mention that line of work, and fit into it where possible.)
> > >
> > > > The final question is really insightful. Since the basic assumption of ad hoc teamwork is that all agents are assumed to have a common goal, so the situation you consider is out of scope.
> > >
> > > I'm not sure I understand. Are you saying that you're assuming that other agents are playing parts of an (welfare-)optimal joint policy? Or that, if all agents run your algorithm, they are guaranteed to converge to a optimal joint policy? (this would be very surprising to me if it were true, as most algorithms usually do not have this property unless explicitly designed for it!) Or that, regardless of the policies of other agents, there is always a policy for the learner such that the resulting joint policy is optimal? Or something else entirely? In any case, the claim here requires explicit statement and justification. But this comment raises yet another question: what does it mean for agents to have a "common goal"? They don't share a reward function, right? Doesn't that mean that their goals may be misaligned?
> > >
> > > This response makes me feel like I am missing something very crucial here. Whatever it may be, though, it certainly requires clarification.
> > >
> > > **3)** (on $\mathcal{CS}_t$) I feel that the importance of clear writing in this paper outweighs the concerns raised by the authors; as such, I'd advise streamlining the notation by removing $\mathcal{CS}_t$.
> > >
> > > **5)** (on types). I understand the general use of having types/partial observability in games. My comment was more to do with the framing/cleanliness of notation of the paper: if the results of the paper is also interesting in a fully-observed setting and the generalization to partial observability is easy enough, maybe it is better to frame the paper mainly for fully-observed setting (because the notation is cleaner) and reserve the extension to partial observability to an appendix. I don't know whether this is possible, but I am raising the suggestion just in case, because anything that would reduce the notation/clarity burden in this paper would be a good thing. If the fully-observed setting is somehow trivial, the authors should point this out.

---

> ### Author Response · Authors · 2023-11-17
> **Further Response to Reviewer eRVD (1/2)**
>
> Thanks for your so fast reply.
> ## Before answering the further questions, we urge the reviewer to calm down when meeting an unfamiliar area during review and give full respect to this area. We are always patient to answer your questions.
>
> **1. (Single-agent perspective) This remark from the authors raises more questions to me than it answers. The assumption that all other agents are playing a fixed policy completely defeats the purpose of investigating a multi-agent system in the first place---you can treat agents with fixed policies as part of the environment and just solve the resulting (single-agent) POMDP. If this paper is really about a single-agent perspective, the authors should do much more to justify why we should believe that this single-agent perspective is relevant to a multi-agent system. (For example, if all agents run your algorithm, what happens? There is a lot of work investigating multi-agent learning dynamics in games, and I believe that the authors should mention that line of work, and fit into it where possible.)**
>
> Reply: The main issue is that **the reviewer is not familiar with ad hoc teamwork**. What this work did follows a topic called ad hoc teamwork which is **totally different** from multi-agent learning. In ad hoc teamwork, the perspective is to control an agent, while in multi-agent learning, the perspective is to control multiple agents. **Also, we have mentioned the difference between ad hoc teamwork and multi-agent learning in Related Works. We hope the reviewer can check the paper carefully before raising questions.** As the reviewer said, the ad hoc teamwork can be regarded as POMDP, however, the key point is that POMDP is a very general model. If the details of such a model is unknown, how should we well solve the further complicated problem you raised that "if all agents run your algorithm"? The setting in the reviewer's question would make environment non-stationary (or time-varying from the perspective of control theory), which is impossible to be solved if we know nothing about the environment. This is the reason why we study the agent modelling first with the assumption of fixed teammates' policies, which we believe is the key step towards the more complicated problems. Does the reviewer agree with this?
>
> **2. I'm not sure I understand. Are you saying that you're assuming that other agents are playing parts of an (welfare-)optimal joint policy? Or that, if all agents run your algorithm, they are guaranteed to converge to a optimal joint policy? (this would be very surprising to me if it were true, as most algorithms usually do not have this property unless explicitly designed for it!) Or that, regardless of the policies of other agents, there is always a policy for the learner such that the resulting joint policy is optimal? Or something else entirely? In any case, the claim here requires explicit statement and justification. But this comment raises yet another question: what does it mean for agents to have a "common goal"? They don't share a reward function, right? Doesn't that mean that their goals may be misaligned?**
>
> Reply: We **never** say that we would tend to find a joint optimal policy or we finally find the optimal joint policy. The learner's policy obtained (combined with other teammates' policies) is almost a sub-optimal joint policy, which is totally different from multi-agent learning (which aims to optimize a joint policy, so that it can find a joint optimal policy). The common goal can be interpreted into two scenarios: all agents share a reward function to describe the common goal, or each agent has a reward function that is the composition of a shared reward to describe the common goal and an individual function that is not conflicting with the common goal (see Section 2.1 in [1]). The key problem in ad hoc teamwork is that even if all agents have the shared reward, it is difficult to make them collaborate if they were never trained in the same environment before. Does it make more sense now?
>
> **3) (on $\mathcal{CS}\_{t}$) I feel that the importance of clear writing in this paper outweighs the concerns raised by the authors; as such, I'd advise streamlining the notation by removing $\mathcal{CS}\_{t}$.**
>
> Reply: We respect the reviewer's viewpoint, however, this is not the critical point to decide the acceptance of the paper.
>
> ## Reference
> [1] Mirsky, Reuth, et al. "A survey of ad hoc teamwork research." European Conference on Multi-Agent Systems. Cham: Springer International Publishing, 2022.

---

> > ### Author Response · Authors · 2023-11-17
> > **Further Response to Reviewer eRVD (2/2)**
> >
> > **(on types). I understand the general use of having types/partial observability in games. My comment was more to do with the framing/cleanliness of notation of the paper: if the results of the paper is also interesting in a fully-observed setting and the generalization to partial observability is easy enough, maybe it is better to frame the paper mainly for fully-observed setting (because the notation is cleaner) and reserve the extension to partial observability to an appendix. I don't know whether this is possible, but I am raising the suggestion just in case, because anything that would reduce the notation/clarity burden in this paper would be a good thing. If the fully-observed setting is somehow trivial, the authors should point this out.**
> >
> > Reply: Thanks for the reviewer's interesting suggestion. If we understand correctly, the word "partial observability" you mean indicates the unknown types. If so, we would say this could be not a good idea. Since in the area of ad hoc teamwork, unknown agent type is a critical point that becomes a non-negligible part when talking about ad hoc teamwork. If following the reviewer's suggestion, we are afraid that this would lose attention of the audience of the area of ad hoc teamwork. However, we understand the reviewer's concerns, which seems reasonable to a general audience. Nevertheless, how to balance between the different groups of audience for a research paper is always a big issue and is still an open question. Does the reply make sense?

---

> ### Comment · Reviewer_eRVD · 2023-11-18
>
> Thank you for the fast response as well. I am willing to engage in discussion about this paper, as should be evidenced by my own fast replies here. As I did say in my review (and you have definitely noticed), this specific area is not very familiar to me. I also cannot become a topic-area expert in a matter of a few days. That said, I do work more broadly in game theory, so I would think that I am at least somewhat in the target audience, enough that I should be able to understand the broad takeaways and motivations of the paper.
>
> I think I am understanding a bit better, but still missing something very crucial. I hope this time though my comments are more pertinent.
>
> Theorem 1 states that the *optimal joint policy* exhibits strict core stability. So, since you're not finding an optimal joint policy in any sense, Theorem 1 does not apply to your algorithm. Now, assuming I've understood this part correctly, I ask/recommend:
>
> 1. I think the above should be explicitly stated in the paper immediately after the statement of Theorem 1, and also in the same place it should be explained why Theorem 1 is even present/relevant to the paper, given that it does not apply to your main algorithm. Otherwise a reader could easily assume (as I did!) that the paper will want to somehow achieve a joint optimal policy, or something like it, in order to apply Theorem 1, and get confused/fixated.
> 1. The paper very quickly moves from jointly optimizing the social welfare (Theorem 1/Corollary 1) to a single-agent optimization problem (Eq. (3)). The relationship between these should be explained, since solving Eq (3) does not in itself guarantee welfare optimality. This may be related to the previous point.
> 1. Which theoretical results of the paper *do* apply to your algorithm? Can you state a concrete result along the lines of "if we run CA-GPL for one agent, with other agents' policies fixed and arbitrary, we will achieve strict core stability"? If so, such a result should be explicitly stated. If not, you should explain what guarantees *are* achieved by your algorithm, because right now the theoretical results are about strict core stability in general and it is not clear how they relate to your algorithm.
> 1. For what joint policies do the results of Section 4.2 hold? It looks to me like those results should hold for *all* joint policies, but if so, that should be explicitly stated. And, in that case, I ask what Section 4.2 has to do with learning a good policy at all, because if all joint policies satisfy the desired property then the satisfaction of the property isn't a constraint on the learning at all.
> 1. In Section 4.2, it seems that you are making assumptions about the structure of weights $w$ (nonnegative) and rewards ($R_j(s_t, a_{t, j}) = (1/|{-i}|) R_i(s_t, a_{t, i})$). These assumptions should be justified; they seem quite strong to me.
>
> If I have misunderstood something above, I'd be happy for the authors to correct me. I am pretty sure I am not the only reader who will have these confusions, so I do hope that this conversation leads to a stronger, more broadly understandable writeup.
>
> The reply about types makes sense to me. Thank you for clarifying.

---

> > ### Author Response · Authors · 2023-11-18
> > **Further Response to Reviewer eRVD (1/2)**
> >
> > **1. That said, I do work more broadly in game theory, so I would think that I am at least somewhat in the target audience, enough that I should be able to understand the broad takeaways and motivations of the paper.**
> >
> > Reply: We totally agree with your statement and prospect. This is the reason why we patiently discuss and explain our work with you, which we think is the biggest purpose of openreview system brought up by ICLR. Now, everything seems going towards the positive direction.
> >
> > **2. I think the above should be explicitly stated in the paper immediately after the statement of Theorem 1, and also in the same place it should be explained why Theorem 1 is even present/relevant to the paper, given that it does not apply to your main algorithm. Otherwise a reader could easily assume (as I did!) that the paper will want to somehow achieve a joint optimal policy, or something like it, in order to apply Theorem 1, and get confused/fixated.**
> >
> > Reply: Yes, we agree with this point and are willing to revise it in the next version when we reach concensus for the question below.
> >
> > **3. The paper very quickly moves from jointly optimizing the social welfare (Theorem 1/Corollary 1) to a single-agent optimization problem (Eq. (3)). The relationship between these should be explained, since solving Eq (3) does not in itself guarantee welfare optimality. This may be related to the previous point.**
> >
> > Reply: We understand the reviewer's concerns. We now clarify the relationship between jointly optimizing the social welfare and the single-agent optimization problem. If we apply multi-agent reinforcement learning, it is easy to reach the optimal social welfare with deciding all agents' actions. Nevertheless, in the context of ad hoc teamwork, the restriction of controlling only one agent (the learner) would impede the convergence to the optimal social welfare.
> >
> > To address this problem, when we solve the single-agent optimization problem, we also need to know other teammates' types and policies, denoted as $P_{E}$ and $\pi_{t,-i}$ in Eq. (4).
> >
> > However, $P_{E}$ is unknown and it is estimated in practice following the result of Proposition 3 (see Section Further Implementation Details in Appendix). The teammates' policies are then estimated following the affinity graph structure as a star graph. To verify the assumption of graph structure as a star graph for policy learning (in contrary to the fully connected graph in GPL), we did an experiment in Section 5.2 Graph Structure to Aggregate Observations for Teammates’ Policies. These are the same architecture as GPL, but we given them a full explanation from theoretical perspective in our work that was missing in the GPL paper.
> >
> > Based on the above estimation, we can get the model of environment (which is constituted of environmental dynamics and teammates' model) to reach the social welfare as optimal as possible.
> >
> > **4. Which theoretical results of the paper do apply to your algorithm? Can you state a concrete result along the lines of "if we run CA-GPL for one agent, with other agents' policies fixed and arbitrary, we will achieve strict core stability"? If so, such a result should be explicitly stated. If not, you should explain what guarantees are achieved by your algorithm, because right now the theoretical results are about strict core stability in general and it is not clear how they relate to your algorithm.**
> >
> > Reply: Following our explanation above, Theorem 1 introduces the optimization problem we use, along with using a graph to describe the teammate model. Theorem 2 and Proposition 2 suggest the range of pairwise and individual utilities as greater than or equal to zero during implementation. If we satify the results from Theorem 1 (optimizing social welfare) and Theorem 2 (confining the weights of the learned affinity graph), then it is possible for us to reach the strict core stability. This depends on how the estimation is accurate and how teammates' desire towards collaboration in practice. Is it clearer now?

---

> > > ### Author Response · Authors · 2023-11-18
> > > **Further Response to Reviewer eRVD (2/2)**
> > >
> > > **5. For what joint policies do the results of Section 4.2 hold? It looks to me like those results should hold for all joint policies, but if so, that should be explicitly stated. And, in that case, I ask what Section 4.2 has to do with learning a good policy at all, because if all joint policies satisfy the desired property then the satisfaction of the property isn't a constraint on the learning at all.**
> > >
> > > Reply: For the first part of question, the results of Section 4.2 hold for the teammates' policies that satisfy the assumption of ad hoc teamwork, which are designed for (learned from) a shared reward function or the reward function composed of a shared reward function and an individual reward function not conflicting with the common goal.
> > >
> > > For the second part of question, if we understand your question correctly, **the takeaway of Section 4.2 is that the pairwise and individual utilities should be implemented with a regularization to confine the range of their outputs to be non-negative**, to facilitate the learning. These are mentioned in the text below Proposition 2. We also validate our method empirically in experiments by comparing CA-GPL with its variants with the ranges that are opposite to our conclusion (e.g., negative or zero).
> > >
> > > **6. In Section 4.2, it seems that you are making assumptions about the structure of weights $w$ (nonnegative) and rewards ($R_{j}(s_{t}, a_{t,j}) = \frac{1}{|-i|} \cdot R_{i}(s_{t}, a_{t,i})$). These assumptions should be justified; they seem quite strong to me.**
> > >
> > > Reply: About $R_{j}(s_{t}, a_{t,j}) = \frac{1}{|-i|} \cdot R_{i}(s_{t}, a_{t,i})$, it is difficult to say whether this condition is strong or not. The main reason is that this condition is highly related to each agent's own property, the study of which is blurry in the existing literature. We will mention it in the next version of revised paper. What we agree is that this is difficult to be interpreted for now and it is necessary to be studied in details in the future work. In our view, the development of research should be progressive, step by step.
> > >
> > > As for the structure of weights $w$ (nonnegative), it is not an assumption, instead it is a range derived from our theoretical model, where the solution of strict core stability should lie in. This guides us to implement the pairwise and individual utilities with a reasonable regularization (using an absolute function as the activation functions of the outputs), which was never mentioned in the GPL paper. However, this result would lead to the flaw that the global reward needs to be nonnegative also. For this reason, we have raised a discussion about it in Section 4.4. The takeaway is that this would not be a problem to influence the learning in implementation.
> > >
> > > **Finally, we would say the reviewer's attitude in delving into details worths recognition. We have to admit that some of your questions are really insightful and not easy to answer. We are happy to make the reviewer understand our work better.**

---

> ### Comment · Reviewer_eRVD · 2023-11-18
>
> (Note: You've numbered your responses off-by-one from my questions. For continuity, I'll switch to your numbering.)
>
> **3-4)** This answer is somewhat disturbing to me. Are you saying that, without further conditions (e.g., on how other agents are reacting to the learning agent), *none* of the theoretical results stated in the paper apply to the main algorithm? (Because the algorithm cannot guarantee welfare optimality, which is required for Theorem 1 and therefore according to your response is required in order to say anything about your algorithm).
>
> A previous comment in this thread stated very clearly that this paper is *not* claiming to find a welfare-optimal joint policy, which I agree with. But yet the theory seems to depend on an optimal joint policy being found. This seems to be a fundamental disconnect here between the algorithm and the theory that severely limits the relevance of the theoretical results.
>
> **5)** What, formally, is "the assumption of ad-hoc teamwork"? Is it a condition on the *rewards*/affiinity graph structure in addition to the structure in Section 3.1 and 3.2? A condition on the *training procedure* (Not sure what precisely that would entail, should specify)?  A condition on the *learned policies* (e.g. that the joint policy is welfare-optimal, as in Theorem 1)? Theorem 2's statement and proof doesn't seem to include any such additional assumption, so if there is an additional assumption it should be explicitly stated.
>
> **6)** Okay, seems like I've misunderstood something here. Are you saying that *your algorithm* is allowed to *change the weights* $w$? Section 3.1 makes it look like $w$ is part of the *input*/*problem definition*, so you can't change them as you see fit to achieve a desirable result. If the algorithm is allowed to set the weights $w$, this should be explicitly stated; it is yet another rather strong assumption that the paper seems to be implicitly making.
>
> Actually, regarding the reward assumption, another question: does it have to hold for *every* $s_t, a_{t, i}, a_{t, j}$? Or only some? If only some, which ones? If every, I think we have a problem:
> 1. By varying $a_{t, j}$, one immediately concludes that every action for every player must have the same reward, i.e., that the reward must be independent of the action.
> 1. If there are $n>2$ agents, then applying the condition twice yields $R_j(s_t) = R_i(s_t)/(n-1) = R_j(s_t)/(n-1)^2$ (where I omit the actions because, by the previous point, the reward must be action-independent) but the only reward function that satisfies this is the reward function that is identically zero, which is obviously not very interesting.
> Am I  missing something here?

---

> > ### Author Response · Authors · 2023-11-18
> > **Further Response to Reviewer eRVD (1/2)**
> >
> > **3-4) This answer is somewhat disturbing to me. Are you saying that, without further conditions (e.g., on how other agents are reacting to the learning agent), none of the theoretical results stated in the paper apply to the main algorithm? (Because the algorithm cannot guarantee welfare optimality, which is required for Theorem 1 and therefore according to your response is required in order to say anything about your algorithm). A previous comment in this thread stated very clearly that this paper is not claiming to find a welfare-optimal joint policy, which I agree with. But yet the theory seems to depend on an optimal joint policy being found. This seems to be a fundamental disconnect here between the algorithm and the theory that severely limits the relevance of the theoretical results.**
> >
> > Reply: Yes, the theory depends on the optimal joint policy of other teammates being found. Note that teammates' policies can react the learner's action (e.g. the learner's action is part of their observations). This is the reason why the learner is required to generate actions that influence teammates (where the learner's policy is learned by Eq. (4) which is a direct construction resulting from Theorem 1) to perform actions as optimal as possible (if these teammates' optimal actions can be generated from their policies). Therefore, the only gap between Theorem 1 and the resulting algorithm is whether the teammates' optimal actions can be generated under the influence of the learner. In our view, this does not mean that the relevance of our theoretical results and the algorithm is limited.
> >
> > **5) What, formally, is "the assumption of ad-hoc teamwork"? Is it a condition on the rewards/affiinity graph structure in addition to the structure in Section 3.1 and 3.2? A condition on the training procedure (Not sure what precisely that would entail, should specify)? A condition on the learned policies (e.g. that the joint policy is welfare-optimal, as in Theorem 1)? Theorem 2's statement and proof doesn't seem to include any such additional assumption, so if there is an additional assumption it should be explicitly stated.**
> >
> > Reply: The answer to "Is it a condition on the rewards/affiinity graph structure in addition to the structure in Section 3.1 and 3.2?" is no.
> >
> > The answer to "Theorem 2's statement and proof doesn't seem to include any such additional assumption, so if there is an additional assumption it should be explicitly stated." is that no additional assumptions.
> >
> > The answer to "A condition on the training procedure (Not sure what precisely that would entail, should specify)? A condition on the learned policies (e.g. that the joint policy is welfare-optimal, as in Theorem 1)?" is as follows. First, we would like to mention that we consider the open team ad hoc teamwork in this paper. If you mean the condition that "It is undefined when $\mathcal{N}_{t} \subset \mathcal{N}_{t+1}$", then the answer is that since generally Q-value is an estimation of an agent's all future decisions, in our case it is unreasonable to expand the joint Q-value at timestep t, where a teammate does not appear in the team at timestep t, but it appears in the team at timestep t+1. If so, this agent would be most likely a ghost. :)
> >
> > The key formal assumptions of ad hoc teamwork are as follows:
> > 1. The learner is expected to cooperate with its teammates when the task begins without any prior opportunities to establish or specify mechanisms for coordination.
> > 2. The learner cannot change the properties of the environment, and the teammates’ policies and communication protocols; it has to reason and act under the given conditions.
> > 3. All agents are assumed to have a common objective, but some teammates might have additional, individual objectives, or even completely different rewards. However, these additional objectives do not conflict with the common task (Grosz and Kraus 1999).
> >
> > **6) Okay, seems like I've misunderstood something here. Are you saying that your algorithm is allowed to change the weights $w$? Section 3.1 makes it look like $w$ is part of the input/problem definition, so you can't change them as you see fit to achieve a desirable result. If the algorithm is allowed to set the weights $w$, this should be explicitly stated; it is yet another rather strong assumption that the paper seems to be implicitly making.**
> >
> > Reply: Yes, our algorithm is allowed to change the weights $w$. More exactly, to follow the paradigm of reinforcement learning, we estimate the cumulative weights, denoted by $Q_{j}$ that is composed of $Q_{ij}$, $Q_{i}$ and $Q_{j}$. This is not a strong assumption, instead, it is an online estimation of weights given the interaction between the learner and teamates, which is what Eq. (5) does. The whole logic of our paper is that we firstly give theorems with all necessary conditions. Due to that some conditions cannot be directly satisfied, we use learning method to reach that (data driven).

---

> > > ### Author Response · Authors · 2023-11-18
> > > **Further Response to Reviewer eRVD (2/2)**
> > >
> > > **Actually, regarding the reward assumption, another question: does it have to hold for every $s_{t}, a_{t,i}, a_{t,j}$? Or only some? If only some, which ones? If every, I think we have a problem:**
> > >
> > > Reply: Yes, for now we assume that it is for every. However, we guess that the reviewer only discusses the reward of each agent where only itself forms a coalition, e.g., $R_{j}(s_{t}, a_{t,j})$.
> > >
> > > **1. By varying $a_{t,j}$, one immediately concludes that every action for every player must have the same reward, i.e., that the reward must be independent of the action.**
> > >
> > > **2. If there are $n > 2$ agents, then applying the condition twice yields $R_{j}(s_{t}) = R_{i}(s_{t})/(n-1) = R_{j}(s_{t})/(n-1)^{2}$ (where I omit the actions because, by the previous point, the reward must be action-independent) but the only reward function that satisfies this is the reward function that is identically zero, which is obviously not very interesting. Am I missing something here?**
> > >
> > > Reply: Yes, your analysis should be correct. Thanks for giving us an insight into the $R_{j}(s_{t}, a_{t,j})$. Although the result seems not very interesting, it instead simplify the understanding of Proposition 2. Following your idea, if discussing the relation between two actions, a very interesting result could be turned out. This could be left to the future work.
> > >
> > > **Thanks for the reviewer's suggestion and discussion. We really enjoy the whole process and sincerely hope our explanation can help you understand our work better.**

---

> ### Comment · Reviewer_eRVD · 2023-11-19
>
> **Regarding Theorem 1 and joint (welfare) optimality:** I think you are saying that the "only gap" in the application of Theorem 1 is that the joint optimal policy may not be reached. But, in an earlier message, you stated very clearly that your algorithm does not in any way claim to allow the joint optimal policy to be reached. So, this "gap" is not a small gap at all; rather, it is a fundamental limitation of the algorithm that prevents the application of the main theorem!
>
> To me, this gap is a severe limitation that undermines not only the relevance of Theorem 1, but also the relevance of the whole approach of the paper since, if I understand the authors correctly, all the theoretical results about the main algorithm depend on joint optimality.
>
> Also, if there is some way to formalize what it means to "influence" the other agents, what conditions are being implied here about how the other agents are learning/reacting to the learner's play, and what that means about the eventual joint policy of the players, that would be very important to include. But it is not really directly relevant to my main point above; that point only relies on the authors' own statements that:
> 1. the main algorithm is not claiming to reach a joint optimal policy.
> 2. positive results about the main algorithm depend on reaching a joint optimal policy ("If we satify the results from Theorem 1 (optimizing social welfare) and Theorem 2 (confining the weights of the learned affinity graph), then it is possible for us to reach the strict core stability.")
>
>  Am I missing something?
>
> **Regarding being allowed to change weights:** Section 3 is written in such a way that it is implied that the weights are part of the input, i.e., they are not things that are learned by the algorithm. Section 3 should be rewritten to fix this.
>
> **Regarding the reward assumption:** I see. So, under the assumptions of that section, singleton coalitions are assumed to have reward zero when $n>2$, but non-singleton coalitions can still easily have nonzero reward. Both these points should be stated explicitly in the text.

---

> > ### Author Response · Authors · 2023-11-19
> > **Further Response to Reviewer eRVD**
> >
> > **Regarding Theorem 1 and joint (welfare) optimality: I think you are saying that the "only gap" in the application of Theorem 1 is that the joint optimal policy may not be reached. But, in an earlier message, you stated very clearly that your algorithm does not in any way claim to allow the joint optimal policy to be reached. So, this "gap" is not a small gap at all; rather, it is a fundamental limitation of the algorithm that prevents the application of the main theorem!**
> >
> > Reply: In our view, it is meaningless to stick on the word game and there may exist some language barrier. However, the these two sentences are actually consistent. Since we only formulate a theory to model an unknown situation (who knows whether the agents in realistic really satisfy any properties) based on our assumption, it is highly likely there exists a gap. For this reason, the main purpose of the work is not rigorously verify any Theorem (if so, we have to guarantee all conditions have to be satisfied), which is the meaning of "we would not (100\%) be able to reach the the joint optimal policy". To show the correctness of our assumption and theory, the best way is to verify the performance in comparatively complicated experiments. In other words, the main purpose of this work is to model ad hoc teamwork as the title shows, so the experimental result is the best evidence to verify its correctness.
> >
> >
> > **To me, this gap is a severe limitation that undermines not only the relevance of Theorem 1, but also the relevance of the whole approach of the paper since, if I understand the authors correctly, all the theoretical results about the main algorithm depend on joint optimality.**
> >
> > Reply: This is the reason why in implementation we try to track the behaviour of uncontrollable teammates. This also happens in control theory, using state estimation to track the uncertain terms in theory. However, what they further do is that they also provide a proof of the process they can accurately track the uncertain term. Similarly, in our work, what we do is about the first step, and then propose to track the uncertainty with an empirical method. What we would like to argue is that this is only a paper of 9 pages, it is impossible to include all possible things. In the future work, we would further investigate the convergence of algorithm under the online estimation.
> >
> > **Also, if there is some way to formalize what it means to "influence" the other agents, what conditions are being implied here about how the other agents are learning/reacting to the learner's play, and what that means about the eventual joint policy of the players, that would be very important to include.**
> >
> > Reply: Yes, we agree. This is our next step, but this can only be forwarded when this paper is accepted, since we need to rely on the model proposed in this work.
> >
> > **But it is not really directly relevant to my main point above; that point only relies on the authors' own statements that: the main algorithm is not claiming to reach a joint optimal policy. positive results about the main algorithm depend on reaching a joint optimal policy ("If we satify the results from Theorem 1 (optimizing social welfare) and Theorem 2 (confining the weights of the learned affinity graph), then it is possible for us to reach the strict core stability.")
> > Am I missing something?**
> >
> > Reply: Your understanding is generally correct. However, it could be helpful to rearrange the logic. Theorem 1 -> objective function based on $P_{E}, \pi_{t,-i}$; Theorem 2 -> confining the search space of learnable variables; Theorem 3 (along with Eq. (5) and (6)) -> Introduce the learning method to find the learnable variable; Proposition 3 -> to introduce an empirical estimator to estimate $P_{E}$ (along with $\pi_{t,-i}$ following the paradigm of GPL). Again, we would argue that Theorem 1 is a kind of key to introduce the following methods, which is called theoretically guided. This manner is popular in machine learning. Of course, during discussion we can feel your background and research flavor, which is different from ours. However, the most important is that we should show more undrstanding to each other, rather than merely judging from our own sides, which can help improve the research field better.
> >
> > **Thanks for your dedicated work again.**

---

> ### Comment · Reviewer_eRVD · 2023-11-19
>
> I think we will agree to disagree here. Here's another way of phrasing where I am stuck: this work is introducing a new framework based on cooperative game theory to achieve ad-hoc teamwork. But, in some sense, one of the main premises of ad-hoc teamwork *is* that you can't expect to achieve the joint optimal policy because you don't control what the other agents are doing (otherwise, it'd be a MARL problem). So the field attempts to understand what *is* achievable in this setting when you have so little control. I understand that motivation. But, with that in mind, developing a framework whose entire conceptual/theoretical logic is "if we achieve the thing that we just said is not a reasonable expectation (joint optimality), then we achieve a good outcome (strict core stability)" now seems almost contradictory to the above premise.
>
> I would strongly suggest that the authors revisit the fundamental conceptual premises of the paper, without assuming joint optimality. There could still definitely be interesting directions to investigate at the intersection of cooperative game theory and ad-hoc teamwork. For example, there could be other interesting statements that can be made using cooperative game theory that do not assume joint optimality. Or perhaps, one could experimentally check whether your joint policies exhibit nice properties from cooperative game theory (such as, but perhaps not limited to, core stability), even if they are suboptimal? Of course, I cannot expect this to be done in what remains of the discussion period, so these are suggestions for a future revision.
>
> > Reply: Your understanding is generally correct. However, it could be helpful to rearrange the logic.
>
> The paragraph that follows this sentence was not obvious to me. The information/intuition in that paragraph should be explicitly discussed in a revision.
>
> I think I will not have much more significant or new things to say at this point, and I have spent already a very large amount of time on this paper, so this will be my last message here and I will maintain my score. Thank you for being engaging in the review process. I hope that the discussion has been helpful.

---

> > ### Author Response · Authors · 2023-11-19
> > **Further Response to Reviewer eRVD**
> >
> > **But, with that in mind, developing a framework whose entire conceptual/theoretical logic is "if we achieve the thing that we just said is not a reasonable expectation (joint optimality), then we achieve a good outcome (strict core stability)" now seems almost contradictory to the above premise.**
> >
> > Reply: The objective is "if we achieve the thing that we just said is not a reasonable expectation (joint optimality), then we achieve a good outcome (strict core stability)". **Then, what we do is to enable the learner to influence other agents to achieve the optimal joint policy as much as possible, so as to achieve a good outcome. (We have repeated several times)**
> >
> > ## The above logic is APPARENTLY NOT contradictory. :)
> >
> > ## This would be also our last message, and we also have spent a lot of time to explain some common knowledge to you. Thank you again or engaging in the discussion. Hope you enjoy the discussion. Have a good day. :)

---

### Official Review · Reviewer_23zx · 2023-10-25

**Soundness:** 3 good
**Presentation:** 2 fair
**Contribution:** 3 good
**Rating:** 5
**Confidence:** 3

**Summary:**

The paper studies the problem of ad-hoc teamwork from a novel, cooperative game-theory perspective. Specifically, the paper considers conditions on the optimization of the single learner (in ad-hoc teamwork, a single learner is trained to collaborate with different teammates which she has never met before) so that the grand coalition, i.e., the coalition in which all teammates collaborate for the common goal, is stable and preferred by all agents. Importantly, this allows for an approach with variable number of teammates in every round that is not addressed by previous methods. The paper further uses a, possibly time-varying, star graph to model the interactions of the players with the learner in the centre and the non-interacting teammates in the leaves.

The main contribution of the paper is that it proves that the solution concept describing the stability of ad hoc teams, roughly the core of the cooperative game described above, is reached when the agents maximize the social welfare, i.e., the sum of agents’ preference Q-values as the global Q-value. Based on this, the paper extends the Graph-based Policy Learning (GPL) algorithm to the coalition-affinity GPL (CA-GPL). CA-GPL is then experimentally evaluated in the Level-Based Foraging (LBF) and Wolfpack environments. The experiments suggest consistent improvements over GPL and also dilligently highlight the importance of each assumption used in the theoretical results. For instance, variations of the main model that relax sufficient (not necessary) conditions show similar performance to the optimal model sheding light into possibilities for further research.

**Strengths:**

- The paper is well placed in the relevant literature of ad-hoc teamwork. It considers a SoTA algorithm, identifies well-justified shortcomings, e.g., lack of theoretical foundation/suboptimal results in variable environments, and tries to propose solutions to these.
- The paper is using novel techniques, specifically elements from cooperative game-theory, to provide a theoretical background for a simple algorithm that provides improvements on the above problems.
- The environments used in the experiments and the experimental evaluation are comprehensive, include sensitivity (ablation) studies of all assumptions required for the theoretical results and are very well presented. The paper also provides a link to a website that includes demos of the experiments which provides further insight. I am not sure, though, if links are desirable or if these demos need to be provided in the supplementary material to make the paper self-contained.
- Limitations, proofs and simulation details for reproducibility are thoroughly documented.

**Weaknesses:**

- The major weakness of the paper is, in my subjective opinion, the overly complicated presentation in some critical places which does not allow the reader to appreciate the theoretical contributions of the paper and to verify their correctness. This affects both the theory and the experiments and I elaborate on this below in the experiments.

In general, I think that the paper achieves a solid contribution in the literature and has a clear potential, but given my concerns about proper understanding, I think that it requires a thorough revision prior to being ready for publication.

**Questions:**

I elaborate below on my comment in the weaknesses:
- Can you please elaborate on what do types represent since all teammates always want to collaborate? This is not explained in the text.
- Can you please highlight the main theorem that addresses the motivating question at the bottom of page and top of page 2? I admit that I got lost in some parts of the theoretical presentation.
- An algorithm environment for CA-GPL would have aided the reader to appeciate the novelties over GPL.
- If I am not mistaken, the notation seems too complicated and redundant. For instance, if I understand correctly, it holds that v_j(C) = w_ji  = R_j and later on R_j = \alpha_j + (other terms). Similarly for the social welfare that is defined as sum (of sums) of such terms. Thus, I had difficulty to follow the proofs. Is my concern valid?
- More examples for complicated notation:
    - what is $|-i|$? Is it the cardinality of the set of all agents other than $i$? If yes, then first, this is never explained, and second why not simply write $N_t-1$?
    - what is $b_j$ in the bottom of page 2? Is this used later on?
    - what is $a_{t,j}$ below equation (1)? Should this be $a_{t,-i}$?
    - social welfare is used in Theorem 1, but it has not been defined before.
- It seems to me that the definition of the joint policy $\pi_{t,-i} :\mathcal{S}\times \Theta_N \to \Delta(\mathcal{A}_N)$ allows for _correlated_ policies. However, this contradicts the assumption that permeats the text, that teammates don't interact at all with each other. Is this concern valid?
- Theorem 1 is hard to parse for me. If the grand coalition is strict core stable (with respect to what valuations?), then what does it mean to show that it is strict core stable? Apologies if my confusion is not justified.
- Regarding the experiments: In Table 1, we see that the stability metric worsens for CA-GPL in both scenarios as training progresses (especially for the 4 teammates). Why is this? Also, in the figures, we see that the curves keep increasing (they have not converged). Is this a valid concern? And would it make sense to provide larger frames?

There are many typos or difficult to understand sentences. I name a few below:
- page 2: "we translate the achievement .... decision making". incomprehensible
- p2: the stability, of ad hoc teams (misplaced comma)
- p2: LBF -> please name it properly the first time that this is used.
- p3: "if a coalition structure ... weakly blocking coalition". incomprehensible
- p3: "There exist an affinity graph" -> exists
- p3: continually (you may want to check if continuously is better here)
- p5: we aims at -> aim
- p8: "the relationship among ad hoc teammates is weak to" -> shouldn't it be the opposite here, i.e., strong relationship?
- p9: "theoretical model Our theoretical model" -> typo?
- p16: "...results in that prices ..." (in Theorem 1) -> incomprehensible.
- there are more, please check.

---

> ### Author Response · Authors · 2023-11-17
> **Response to Reviewer 23zx (1/2)**
>
> 1. **Can you please elaborate on what do types represent since all teammates always want to collaborate? This is not explained in the text.**
>
>     *Reply:* The types here is for emphasizing the heterogeneity of agents. Even if all teammates always want to collaborate, a group of different types of agents may collaborate in different manners. Thereby, in this paper we apply affinity graph with learnable weights (capture variant collaborative manner) to depict this property.
>
> 2. **Can you please highlight the main theorem that addresses the motivating question at the bottom of page and top of page 2? I admit that I got lost in some parts of the theoretical presentation.**
>
>     *Reply:* Sorry for making you lost owing to our presentation. For convenience, we will reply following the layout of the revised paper. The answer to the motivating question is constituted of the holistic theory. It starts from the introduction of CAG with the affinity graph and the solution concept for solving the weights of the affinity graph called strict core stability (see Definition 1). Then, we define the affinity graph as a star graph to better describe the scenario of open ad hoc teamwork (see Definition 2). After that, we derive a method to find the solution of weights (see Theorem 1 and the derived Eq. (4)) and an inductive bias for easily finding the weights (see Theorem 2 and Proposition 2). Finally, we show the feasibility of solving Eq. (4) under the open team scenario (see Lemma 1 and Theorem 3). In summary, our theory answers the question that why introducing and learning a graph is necessary and feasible to solve open ad hoc teamwork from the theoretical perspective rather than the pure intuition in the GPL paper, so the method derived from our theory is reliable.
>
> 3. **An algorithm environment for CA-GPL would have aided the reader to appeciate the novelties over GPL.**
>
>     *Reply:* What do you mean by an algorithm environment? Could you please give some more information about it? We have added an achitecture of CA-GPL to illustrate the novelty over GPL in the revised paper as Figure 1 shows.
>
> 4. **If I am not mistaken, the notation seems too complicated and redundant. For instance, if I understand correctly, it holds that v_j\(C\) = w_ji = R_j and later on R_j = \alpha_j + (other terms). Similarly for the social welfare that is defined as sum (of sums) of such terms. Thus, I had difficulty to follow the proofs. Is my concern valid?**
>
>     *Reply:* Thanks for your suggestion. Your understanding for our own theory is correct only for the teammates $j$. For learner $i$, the instantiation of $v_{i}(C)$ or $R_{i}$ to ad hoc teamwork is different, while the abstract definition of $v_{i}(C)$ is the same as $v_{j}(C)$ in the most general hedonic game. More specifically, $v_{j}(C)$ is the most general definition over any arbitrary coalition $C$ in hedonic game. In our case, we replace $v_{j}(C)$ by $R_{j}$ to make the symbols consistent with convention in reinforcement learning. Using $w_{ji}$ to specify $v_{j}(C)$ is only valid in CAG which is a subclass of hedonic game. $w_{ji} = R_j$ is only valid when we consider the star graph. If we initially define $w_{ji} = R_j$ with ignoring $v_{j}(C)$, this would make our theory too narrow and less valuable to be extended to more complicated scenarios as we mentioned in Conclusion. Similarly, \alpha_j + (other terms) is an instantiation of $w_{ji}$. In summary, our theory is formulated from abstract (the known general theory) to detailed scenarios (the situation of ad hoc teamwork) step by step, with both concerns of the extension of our theory and inheriting past works. We wish you can understand our thought and are happy to have more discussion on it.
>
> 5. **what is $|-i|$? Is it the cardinality of the set of all agents other than $i$? If yes, then first, this is never explained, and second why not simply write $N_{t} - 1$?**
>
>     *Reply:* Yes, thanks for the suggestion. We do not write it as $N_{t} - 1$ is for the conciseness in expression.
>
> 6. **what is $b_{j}$ in the bottom of page 2? Is this used later on?**
>
>     *Reply:* $b_{j}$ was originally used to indicate that the singleton coalition value is a constant value greater than or equal to zero. However, we agree with your suggestion and modify it as $v_{j}(\mathcal{C}) = v_{j}(\{j\}) \geq 0$ in the revised paper.

---

> > ### Author Response · Authors · 2023-11-17
> > **Response to Reviewer 23zx (2/2)**
> >
> > 7. **what is $a_{t,j}$ below equation (1)? Should this be $a_{t,-i}$?**
> >
> >     *Reply:* Yes, you are correct. We have fixed it in the revised paper. Thanks for the careful inspection.
> >
> > 8. **social welfare is used in Theorem 1, but it has not been defined before.**
> >
> >     *Reply:* We now refine the contents in this paragraph and social welfare is defined in the revised paper.
> >
> > 9. **It seems to me that the definition of the joint policy $\pi_{t, -i}: \mathcal{S} \times \Theta_{\scriptscriptstyle\mathcal{N}} \rightarrow \Delta(\mathcal{A}_{\scriptscriptstyle{\mathcal{N}}})$ allows for correlated policies. However, this contradicts the assumption that permeats the text, that teammates don't interact at all with each other. Is this concern valid?**
> >
> >     *Reply:* Thanks for the really good question. The correlated policy is not allowed in our work. The definition here is the one we refer to from the origianl GPL paper. If combined with our specified definition in Section 3.1 such that the corresponding coalitional action decided by $\pi_{t, {\scriptscriptstyle \mathcal{C}}} = \times_{j \in {\mathcal{C}}} \ \pi_{j}$, this ambiguity can be addressed. For instance, let $\mathcal{C} = -i$, then $\pi_{t,-i} = \times_{j \in -i} \ \pi_{j}$.
> >
> > 10. **Theorem 1 is hard to parse for me. If the grand coalition is strict core stable (with respect to what valuations?), then what does it mean to show that it is strict core stable? Apologies if my confusion is not justified.**
> >
> >     *Reply:* We believe any concern is necessary to be addressed before publication. To highlight the valuation of strict core stability, we have specified it in an exclusive definition as Definition 1 in the revised paper. We say that a blocking coalition $\mathcal{C}$ weakly blocks a coalition structure $\mathcal{CS}$ if every agent $j \in \mathcal{C}$ weekly prefers $\mathcal{C}$ to $\mathcal{CS}(j)$ and there exists at least one agent $j' \in \mathcal{C}$ who strictly prefers $\mathcal{C}$ to $\mathcal{CS}(j)$. A coalition structure admitting no weakly blocking coalition $\mathcal{C} \subseteq \mathcal{N}$, exhibits the strict core stability.
> >
> >     The implication of Theorem 1 is as follows: If there exist weights ($w_{jj'}$) to guarantee the strict core stability of the grand coalition, then maximizing social welfare is an approach to find the solution. Our previous description of Theorem 1 does not capture our intension well and make confusion to you. We are so sorry about it. In the revised paper, we have removed "the grand coalition is assumed to be strict core stable" without changing the purpose of Theorem 1, to avoid confusion.
> >
> >     Theorem 2 mainly focuses on confining the space of $w_{jj'}$ (as a inductive bias) to facilitate the search of solution. Note that even if the condition in Theorem 2 can guarantee the existence of a grand coalition as a strict core stability solution, we still need to find the appropriate weights (i.e. in long term in the paper for tractability) that satisfy the global reward (which describes the goal of a collaborative task). This justifies the necessity of the result of Theorem 1. We clarify this in the revised paper. We hope our explanation can address your concerns.
> >
> > 11. **Regarding the experiments: In Table 1, we see that the stability metric worsens for CA-GPL in both scenarios as training progresses (especially for the 4 teammates). Why is this? Also, in the figures, we see that the curves keep increasing (they have not converged). Is this a valid concern? And would it make sense to provide larger frames?**
> >
> >     *Reply:* This is really a good question. In general, this verifies the effectiveness of our method on any arbitrary agent-type composition (avoiding the possibility of overfitting). The main reason could be that our star graph architecture with inductive bias on utilities ($Q_{ij} \geq 0, Q_{i} \geq 0, Q_{j} \geq 0$) can capture the relation between the learner and teammates well, without the noisy estimation of relations between teammates that may happen in GPL (which is equipped with a fully connected graph, without any inductive bias on learning utilities). About the curves keeping increasing, it is difficult to be controlled during experiments. Although we agree that this might be resolved with larger frames, due to the page limit it is not easy to provide that in the revised paper.
> >
> > 12. **"the relationship among ad hoc teammates is weak to" -> shouldn't it be the opposite here, i.e., strong relationship?**
> >
> >     *Reply:* We rephrase the sentence as "The comparatively good performance of CA-GPL-FCG could be due to that there still exists relationship among ad hoc teammates in some environments, but not strong enough, so its performance is still worse than CA-GPL" in the revised paper. This is what we would like to express.

---

> > > ### Comment · Reviewer_23zx · 2023-11-20
> > >
> > > I thank the authors for their response. With an algorithm environment, I meant a typical pseudocode environment. The figure is also welcomed, but indeed it requires some description. Overall, it is impossible for me to follow the long discussions about this paper in detail, but, based also on the other reviews, I will maintain my initial recommendation that the presentation of the paper is overly complicated to clear some doubts on its contribution, and to make it assessible by a wider audience and fit for publication at its current state. The positive elements that I see in the paper have been weighed-in in my initial score, thus, I will maintain it.

---

> > > > ### Author Response · Authors · 2023-11-20
> > > > **Response to Reviewer 23zx**
> > > >
> > > > We agree that the pseudocode or the description of figure could help and will add it in the next version. Moreover, we will attempt to make the contribution more comprehensive. Thank you for your service again.

---

### Official Review · Reviewer_UX5a · 2023-10-31

**Soundness:** 2 fair
**Presentation:** 1 poor
**Contribution:** 2 fair
**Rating:** 3
**Confidence:** 4

**Summary:**

The paper investigates learning techniques for **open** ad-hoc teamwork, which is a setting in which an uncontrollably heterogeneous set of agents are put in the same environment, and we want to train a single agent (called learner) to operate in the environment in order to maximize his reward while potentially collaborating with other agents. The other agents are assumed to be of a fixed type (sampled from a distribution), corresponding to a specific policy that is adopted in the environment.
The main challenges that are faced are that the agents cannot leverage any form of pre-coordination and agents have to adapt in real time to a varying number of teammates in the environment.

The paper builds incrementally on a previous approach known as GPL. The GPL technique is a value-based reinforcement learning technique that uses a complex architecture to estimate:
* types of the agents
* joint-action Q-value (decomposed as sum of single action Q-values and pairwise actions Q-values) given types and state
* behavioral model, i.e. the probability of other agents taking specific actions given types and state
Those components are trained together in a RL fashion, by letting the learner pick actions according to a Q-function over its actions, derived from the joint Q-value and the behavioral model.

The contribution of the paper is a different approach to the decomposition of Q-values based on coalitional affinity games. In particular, the paper:
* models the interactions between the agent as a star-shaped pattern around the learner, with symmetric values
* theoretically justifies the optimization of the agent's reward done by GPL as it corresponds to the strict core of the affinity game centered around the learner
* derives a Bellman optimality equation from the star structure
* uses a GPL-like architecture to estimate Q-values of the learner, with the difference that the joint Q-function does not consider pairwise contributions between agents different from the learners

**Strengths:**

The strenghts of the paper can be summarized as follows:
* introduction of a modelling framework that interprets open ad-hoc teamwork as a coalitional affinity game
* slightly better empirical results in terms of value, more stable in terms of stability

**Weaknesses:**

**Severe** lack of clarity throughout the paper:
* when introducing an affinity graph, the values from the singleton coalition are not ported to the new representation, and then they are used when discussing individual rationality.
* in many instances of the paper, assumptions are introduced without justification and they are not properly highlighted (just a "There exitst..."). As an example the existance of an affinity graph is not guaranteed in general as far as I know.
* Many different probability distributions are introduced ($P_E$, $P_O$, $\mathcal T$ ,,,) in Section 3.1, making really obscure the dependencies across variables, Figure 5 from the appendix and a clearer text  may help.
* lack of an example to clarify the settings and its peculiarities.
* the term "Preference Q-Value" is unneeded, as it is just a Q value over the action space of the agent
* the text between Theorem 2 and Propositoin 2 is unreadable and I could not understand it even reading it multiple times. Please rephrase it and provide proper space for the mathematical equations used.
* the GPL framework is never defined, and it could really help to highlight the contributions of the paper.

The usefulness of the theoretical contributions is debatable. In particular, the whole coalitional affinity games(CAG) framework seems like a very elaborate way of justifying the star-shaped Q-value computation novelty introduced, However, such a conclusion derives rather directly from the star-shaped interaction graph between agents, that is assumed.
*   up to theorem 2, the theoretical contributions are just saying that we need to optimize the learner's utility, which coincides with the social welfare thanks to the assumption of a star-shaped graph. Given the context at hand, this result is banal and the introduction of CAG actually makes the formalism uselessly complex

The final architecture developed for the CAG-GPL is not described in detail, nor properly compared with GPL. A picture is needed, as in GPL paper.

Empirical results are difficult to interpret.
* in Figure 1 it seems that CAG-GPL has really similar performance to GPL
* Figure 3 is unreadable: the legend's colors do not match with the plot, and in the Q-value plot there is written "learner" even on the rows not related to the learner

**Questions:**

* Is my interpretation of the architectural differences between GPL and CAG-GPL in the last point of the summary correct? Ie, this paper "uses a GPL-like architecture to estimate Q-values of the learner, with the difference that the joint Q-function does not consider pairwise contributions between agents different from the learners", so only Q_ij, Q_i and Q_j but not Q_jj'

* I could not highlight an important gap between CAG-GPL and GPL apart from training stability. Is there anything I missed in my weakness summary?

* similarly, what is the biggest novelty from a modelling perspective introduced by the CAGs? If possible, can you compare it in terms of differences with GPL?

---

> ### Author Response · Authors · 2023-11-17
> **Response to Reviewer UX5a (1/3)**
>
> 1. **when introducing an affinity graph, the values from the singleton coalition are not ported to the new representation, and then they are used when discussing individual rationality.**
>
>     *Reply:* Thanks for the insightful comment. We were also struggled with the definition of the reward for a singleton coalition. After deliberation, we believe that it is unnecessary to define it with a new representation, since we have defined the reward over any arbitrary coalition in Section 3.1, and the reward for a singleton coalition is just a concrete case.
>
> 2. **in many instances of the paper, assumptions are introduced without justification and they are not properly highlighted (just a "There exitst..."). As an example the existance of an affinity graph is not guaranteed in general as far as I know.**
>
>     *Reply:* Thanks for the interesting comment. "There exists ..." is a type of words to given a boudary, based on which we can analyze the results following logics. As for the existence of affinity graph, we now add discussion on the boundary in the revised paper (See Section 3.2 **Affinity Graph for Ad Hoc Teamwork**). We sincerely hope you could double check it.
>
> 3. **Many different probability distributions are introduced (...) in Section 3.1, making really obscure the dependencies across variables, Figure 5 from the appendix and a clearer text may help.**
>
>     *Reply:* Thanks for your comment. In our view, different probabilistic distributions defined here does not obscure the dependencies across probabilities. Instead, it clarifies the dependencies, since compared with expressing the transition function as a whole probability distribution in SBG, it points out the position of intervention (control) during the whole ad hoc teamwork process, following the view of causal inference. Nonetheless, we agree that a figure could help understanding, we have redrawn Figure 5 in the last version and it is now Figure 4 in the revised paper.
>
> 4. **lack of an example to clarify the settings and its peculiarities.**
>
>     *Reply:* Thanks for the insightful comment. This paper is not like the purpose of popular papers that capture a weakness that a prior work cannot solve and propose a method that can improve the weakness. The research question (motivation) of this paper is that the powerful GPL framework does not have a theoretical understanding of why the incorporation of graph can help. For this reason, we propose to incorporate CAG and surely its theory to mitigate this issue. More importantly, the incorporation of CAG can lead to the possibility the further investigation into more diverse situations (e.g., friend and enermies both exist). Although GPL seemly can address almost every situations, the further improvement of this architecture is a big problem. The main reason is that it does not consider any boudary from theoretical perspective. **However, the question is that we do not know whether the design is the best to any specific situation.** For example, in this paper, we propose to constrain the range of individual and pairwise Q-values to be greater than or equal to zero, according to our theory. Similarly, we can follow the route of the various and well studied research of cooperative game theory to analyze other scenarios (e.g. friend and foe).
>
> 5. **the term 'Preference Q-Value' is unneeded, as it is just a Q value over the action space of the agent**
>
>     *Reply:* Thanks for this interesting point. We use the term "Preference Q-Value" is to credit and respect the work of CAG and cooperative game theory as well as to distinguish our work from others, since the Q-value is derived by the theory over preference value which is the terminology in cooperative game theory. **If following your logic, it is also unnecessary to define Q-value since it is just a function.** We sincerely hope you could reconsider your statement.

---

> > ### Author Response · Authors · 2023-11-17
> > **Response to Reviewer UX5a (2/3)**
> >
> > 6. **the text between Theorem 2 and Propositoin 2 is unreadable and I could not understand it even reading it multiple times. Please rephrase it and provide proper space for the mathematical equations used.**
> >
> >     *Reply:* Thanks for the good point. We agree with your view and have rephrased and rearranged the texts in the revised version of paper. Please check it and give feedback to us that whether it is easier to be understood now.
> >
> > 7. **the GPL framework is never defined, and it could really help to highlight the contributions of the paper.**
> >
> >     *Reply:* Thanks for the useful comment. **Actually, we have defined GPL in details in the paper. Due to the page limits, we leave it in Appendix.** We sincerely hope you can double check it. We now set a pointer to it in the section of practical implementation in the revised paper for clarity.
> >
> > 8. **The usefulness of the theoretical contributions is debatable. In particular, the whole coalitional affinity games(CAG) framework seems like a very elaborate way of justifying the star-shaped Q-value computation novelty introduced, However, such a conclusion derives rather directly from the star-shaped interaction graph between agents, that is assumed. Up to theorem 2, the theoretical contributions are just saying that we need to optimize the learner's utility, which coincides with the social welfare thanks to the assumption of a star-shaped graph. Given the context at hand, this result is banal and the introduction of CAG actually makes the formalism uselessly complex**
> >
> >     *Reply:* Thanks for the thoughtful comment. First, we would like to solve ad hoc teamwork, where the basic requirement is that an agent needs to act to cooperate with other agents (teammates). As we mentioned in the paper (see Section 3.1), the star graph is adequate to solve this problem. We do not agree that more general the result is, more exciting is the result, which should be discussed based on the situation or problem we would like to solve. Additionally, we have also mentioned in Proposition 4 (see Appendix E in the revised paper) that even with the fully connected graph, there still exists a representation of affinity weights to satisfy symmetry, individual rationality and the grand coalition exhibiting strict core stability. **This means that our theory is not limited to the star graph.**
> >
> >     Finally, we **do not** agree that the introduction of CAG makes the formalism uselessly complex. As we mentioned before, the introduction of CAG is only the first step to formalize the graph-based method for ad hoc teamwork (which is also beneficial to the validity of GPL). Then, referring to the complete study in cooperative game theory (see Section Conclusion), some more interesting scenarios such as a situation involving both teammates and enermies can be deeply investigated.
> >
> > 9. **The final architecture developed for the CAG-GPL is not described in detail, nor properly compared with GPL. A picture is needed, as in GPL paper.**
> >
> >     *Reply:* Thanks for the useful suggestion. We have added a figure to describe the architecture in Figure 1 in the revised paper.
> >
> > 10. **in Figure 1 it seems that CAG-GPL has really similar performance to GPL**
> >
> >     *Reply:* Thanks for the useful comment. The similar performance in Figure 1 is mainly due to the simplicity of the scenario (with at most 2 teammates), where the difference in architecture between the fully connected graph (GPL) and the star graph (our method) is limited. For the worst case, GPL would at most overestimate one more edge than CA-GPL. **We belive that this instead verifies the necessity to develop a theory over the star graph as an evidence to support our response to Question 7 above.**

---

> > > ### Author Response · Authors · 2023-11-17
> > > **Response to Reviewer UX5a (3/3)**
> > >
> > > 11. **Figure 3 is unreadable: the legend's colors do not match with the plot, and in the Q-value plot there is written "learner" even on the rows not related to the learner**
> > >
> > >     *Reply:* Thanks for the careful inspection. We have fixed this legend color mismatch typo in the revised paper. Additionally, we are sorry that our initial figure caption and the main text may not have provided a clear explanation. The Q-value plot you referred to presents the Preference-Q (Pref-Q) with the actions illustrated along the x-axis. To clarify, the last row of each heatmap indicates the learner's preference Q-value, while the previous rows depict teammates' Pref-Q, given teammates' actual actions. **Besides, the symbol "tri\_up" denoted as 'learner' marks the current action taken by the learner. By incorporating the learner’s actions into the Q-value plot, our aim is to demonstrate the alignment between the Pref-Q and cooperative behaviors (shown in the top trajectory visulization), even within a variable team setting.** In the heat map, the redder the color, the greater the Q value. It is observable that the learner's actions, marked by "tri_up", are consistently aligned with the areas of highest Q values. Nevertheless, the preference Q-value cannot always perfectly align with the learner's actions due to the learner's inaccurate estimation of teammates' policies. **The most successful alignments between trajectories and Pref-Q justify the efficacy of Pref-Q to encourage collaboration in open ad hoc teamwork and, thus, verify the reasonableness of our theoretical model.** Due to the limitation of page, we have moved Figure 3 to Appendix G.2. Additionally, we have provided a more detailed caption to highlight the interesting findings more effectively. Thanks again for your valuable questions.
> > >
> > > 12. **Is my interpretation of the architectural differences between GPL and CAG-GPL in the last point of the summary correct? Ie, this paper "uses a GPL-like architecture to estimate Q-values of the learner, with the difference that the joint Q-function does not consider pairwise contributions between agents different from the learners", so only Q_ij, Q_i and Q_j but not Q_jj'**
> > >
> > >     *Reply:* Your understanding is correct about the architectural difference between GPL and CAG-GPL. **However, we would argue that this is like a loading question which seems suspicious in ignoring our second contribution that the range of Q_ij, Q_i and Q_j should be greater than or equal to zero.** This is also a key point to improve the performance (see performance of GPL variants). **More significantly, this is the benefit from the introduction of CAG (see Theorem 2 and Proposition 2), which also supports our response to the usefulness of our theory in Question 8.**
> > >
> > > 13. **I could not highlight an important gap between CAG-GPL and GPL apart from training stability. Is there anything I missed in my weakness summary?**
> > >
> > >     *Reply:* Yes, you **missed** the general purpose of introducing CAG and an important configuration of the range of Q_ij, Q_i and Q_j that are greater than or equal to zero which is derived from our theory. We sincerely hope you would carefully check our response to your weakness assessment above.
> > >
> > > 14. **similarly, what is the biggest novelty from a modelling perspective introduced by the CAGs? If possible, can you compare it in terms of differences with GPL?**
> > >
> > >     *Reply:* Please refer to the reponses to Question 3, 4 and 8 for details. In addition, we also explain the implementation of type inference module in GPL in theory rather than **just an intuition** mentioned in the GPL paper, thanks to the definition in Section 3.1 (see Proposition 3). Another biggest novelty is that from our theory, we suggest that the range of Q_ij, Q_i and Q_j should be greater than or equal to zero (implemented in CAG-GPL), which is **impossible** to be found **only by** the architecture proposed by the GPL paper. We sincerely hope our explanation can help you understand our work better.

---

> > > > ### Comment · Reviewer_UX5a · 2023-11-20
> > > >
> > > > I thank the authors for their detailed answer and the modification they made, which go towards what I think is a good direction for the paper.
> > > >
> > > > I will answer in a general form, instead than focusing on a point by point base. I have reviewed the changes to the main body of the paper, but due to time constraint I could not review the ones in the appendices.
> > > >
> > > > * adding the ca-gpl algorithm figure is definitely useful, but without any text accompanying it, it is unclear how to read it (unless people have already read GPL)
> > > > * I keep my opinion that CAGs, as they are currently described in the paper, are just a distraction from the main contributions of the paper, which instead is the use of the concepts of the strict core stability, the assumption of full collaboration among agents, and the star-shaped graph assumption.
> > > >
> > > > My score on the paper remains unchanged; I think that the paper should be structured in a way that clearly highlights how the paper first theoretically builds on top of ad-hoc teamwork (in the RL formalization) by adding the ideas of strict core stability (how about explaining the assumptions of of strict core stability in the RL formalism, adding the comparison with coalitional affinity games in the appendix?). My suggestion is then to go on introduction GPL in the main body, and modifying it as already explained in the paper. Comparison between the figures can be really helpful too.
> > > > In my opinion this would offer a much clearer understanding of the contributions of the paper (and also answer many of the difficulties raised from Reviewer eRVD, which in my opinion is right in his doubts about the paper. Those doubts are to be solved with proper introduction to the topic and to the modeling assumptions used)
> > > >
> > > > Overall, the results and the approach adopted for ad-hoc teamwork are interesting and worth of a publication, but the paper has to be rebuilt in my opinion.

---

> ### Author Response · Authors · 2023-11-20
> **Further Response to Reviewer UX5a**
>
> **We thank the reviewers for response.**
>
> The main purpose of our paper is to propose a general theoretical model that can help improve ad hoc teamwork, **rather than just fix pitfall existing in GPL**. In other words, GPL is just a realization of the algorithm from our theoretical model. For this reason, we do not agree with the reviewer's suggestion.
>
> On the other hand, the purpose of mentioning close relationship to GPL in details in our paper is to show **our respect to the GPL work**. If this instead distracts the main purpose of our paper, we would consider removing it from the main part of paper and just talking about it in Related Works. Thanks for your feedback to consolidate our determination to drop the unworthy part in our current contents.
>
> Finally, to our best knowledge, if our response has addressed your questions or concerns, it should be deserved to raise your score. This is main purpose of your questions from the policy of review: **Think of the things where a response from the author can change your opinion, clarify a confusion or address a limitation. This is important for a productive rebuttal and discussion phase with the authors.**
>
> **We strongly suggest the reviewer to follow the policy in the next review experience, otherwise, it is unfair to the authors' efforts. Please respect everyone's effort and take the resposibility of avoiding the phenomenon of withdrawing after receiving not positive scores which is not helpful to the community. To keep the healthy development of the community is the obligation of every reviewer.**
>
> We sincerely thank you for your effort on the reviewing our work again.

---

> > ### Comment · Reviewer_UX5a · 2023-11-20
> >
> > I think I gave you enough constructive comments on what I see is missing in the paper, and how I would improve the situation.
> > Your answer addressed my questions, and I have a clearer picture of the contributions of the paper. However the paper as it is currently is not avoiding the same doubts that I had, and it still had the weaknesses that I highlighted previously.
> >
> > Therefore my score does not change. I also suggest the authors to do their work without arrogantly reminding me what my work as a reviewer is, which is not constructive at all and just exacerbates the situation. Thanks.

---

> ### Author Response · Authors · 2023-11-20
> **Further Response to Reviewer UX5a**
>
> As a reviewer also, I believe this is constructive to the whole community, whatever you change the score or not (which is your freedom). Thanks.
>
> By the way, we have to mention that your words **"but the paper has to be rebuilt in my opinion."** is one of the arrogant words we have never seen before!

---

> ### Author Response · Authors · 2023-11-21
> **Appology to Reviewer UX5a**
>
> We appologize for our quite aggresive words yesterday, due to our bad mood. Hope you can understand us.
>
> However, we still do not totally agree with your suggestion on emphasizing results (takeaways) through weakening the thinking process. We believe the thought of our paper is more significant than the resulting algorithm. We may have very different philosophy in research, but no one is incorrect.
>
> Nevetheless, we may attempt to find a trade-off between our thoughts in our next revised paper, to convince you (if it is likely that you will be still a reviewer for our paper in the next venue).
>
> Thanks for your service again!

---

### Official Review · Reviewer_XvTp · 2023-11-09

**Soundness:** 3 good
**Presentation:** 2 fair
**Contribution:** 2 fair
**Rating:** 5
**Confidence:** 3

**Summary:**

This work investigates the ad hoc teamwork (AHT) problem. The work introduces coalitional affinity game (CAG) and its Bayesian variant to characterize the AHT problem with open teams. The work further designs the solution concept and an algorithm to tackle the problem. The work mainly compares its performance with graph-based policy learning (GPL). The work even provides an external link to show some gifs about their experimental results.

**Strengths:**

1. This work combines open SBG and CAG to propose open-SB-CA-G. This combination seems natural. The work demonstrates that open CAG is well compatible with CAG.
2. The work subsequently formulate the problem of finding the optimal stationary policy into reinforcement learning. This cast is very natural and it obtains a variant of the Bellman optimality equation.
3. The work provides some lemmas and a significant amount of experimental results.

**Weaknesses:**

Considering that the idea of combining open SBG and CAG is quite natural, and so does the RL formulation, the contribution of the work will concentrate on its practical performance. The amount of testing environments seems a bit lacking, and the difference of the outperformance seems marginal.

**Questions:**

N/A

---

> ### Author Response · Authors · 2023-11-17
> **Response to Reviewer XvTp**
>
> 1. **Considering that the idea of combining open SBG and CAG is quite natural, and so does the RL formulation.**
>
>     *Reply:* Thanks for your very critical comments. However, we would like to argue that the combination of SBG and CAG is not trivial or natural. The primary reason is that SBG is actually following the **non-cooperative game theory** framework, while CAG is under the **cooperative game theory**. These are totally different concepts, where SBG follows the solution concept of Nash equilibrium, whereas OSB-CAG or CAG follows the solution concept of strict core stability. In Section 4, we show the equivalence between these two concepts, then it establishes the connection between SBG and OSB-CAG. Therefore, it not only makes the GPL as an implementation of our framework, but also understands GPL from a theoretical perspective. Note that the original GPL only intuitively (empirically) extends the framework from SBG with a graph to handle open team settings without any theoretical understanding. **More importantly, according to the theory from CAG, we propose to set the range of individual utilities and pairwise utilities as greater than or equal to zero, which was never mentioned in GPL.**
>
> 2. **the difference of the outperformance seems marginal.**
>
>     *Reply:* Thanks for your comments again. If possible, could you please mention the concreate cases where our method only outperforms GPL marginally? In our understanding, CAG-GPL (our method) ourperforms GPL in every scenario, and with a large margin in most of scenarios, which is also recognized by Reviewer eRVD.

---

> ### Comment · Reviewer_XvTp · 2023-11-22
> **Reply**
>
> I thank the authors for the response.
>
> Regarding the margin, I believe Figure 2a, 2b, 2f have almost no margin, and the margin is literally "marginal" in 2c, 2d, 2e. Correct me if you do not believe so.
>
> I also read through other reviews and the discussions. I was a bit surprised that the authors decided to position the work as a "theoretical framework", which seems to be quite ambitious. The presentation did introduce many ambiguities and could this be one of the reasons? It is a natural question how much does this work contributes to *theory*.

---

> > ### Author Response · Authors · 2023-11-22
> > **Further Response to Reviewer XvTp**
> >
> > **Regarding the margin, I believe Figure 2a, 2b, 2f have almost no margin, and the margin is literally "marginal" in 2c, 2d, 2e. Correct me if you do not believe so.**
> >
> > Reply: We respect your understanding of "marginal" and "no margins", but we do not agree with this. What we can emphasize is that the primary algorithm to be compared is GPL. Other baselines are variants of our algorithms to show the correctness of outputs of our theoretical framework.
> >
> > **I also read through other reviews and the discussions. I was a bit surprised that the authors decided to position the work as a "theoretical framework", which seems to be quite ambitious. The presentation did introduce many ambiguities and could this be one of the reasons? It is a natural question how much does this work contributes to theory.**
> >
> > Reply: It depends on your understanding of theory. You have your own right to interpret and justify it. This is the interpretation of theory: "a supposition or a system of ideas intended to explain something, especially one based on general principles independent of the thing to be explained.". Ambiguity may be raised from your own capbility of understanding or our capabilities of description, but it does not change the nature of theory. For this reason, your ironic words seem interesting but definitely incorrect. The final question is very good which we have answered in our last response.

---

### Author Response · Authors · 2023-11-17
**Summary of Main Revision of Paper**

We thank all reviewers for their efforts on reading our paper and providing valudable suggestions.

We improve our paper as per reviewers' comments and advice, where all main revision is shown in red. We believe the revised version is clear enough to be ready for publication. The main revision is highlighted as follows:

1. Rearrange the layout of the whole paper, to make it more comprehensive and easy to find the main contribution.
2. Fix typos appeared in the previous version.
3. Add a flowchart to summarize the architecture of CA-GPL as shown in Figure 1, as suggested by Reviewer 23zx and Reviewer UX5a.
4. Plot Figure 4 to illustrate the probability distributions forming the transition function, as suggested by Reviewer UX5a.
5. Add Remark 1 in the revised paper to highlight the importance of proposing open ad hoc Bellman optimality equation.
6. Refine Theorem 1 to address the concerns of Reviewer 23zx.
7. Add Corollary 1 to further explain the use of Theorem 1 in our theory, as suggested by Reviewer eRVD.
8. Consolidate the boundary of the existence of a star graph, as suggested by Reviewer UX5a.
9. Move the case study to Appendix due to the page limit.

---

> ### Author Response · Authors · 2023-11-17
> **Thanks to Reviewers and ACs**
>
> Dear Reviewers and ACs,
>
> Thank you sincerely for the time and effort you invested in reviewing our paper. Your insightful feedback and constructive suggestions have been invaluable in enhancing the quality and clarity of my work. We have carefully considered and incorporated your recommendations to improve our paper. Additionally, we have tried our best to address your concerns.
>
> We sincerely hope our response can help you understand our work better. If you still have concerns, please raise them without hesitation. We are happy to answer any questions.
>
> Your support and guidance are deeply appreciated, and we are grateful for your contribution to advancing knowledge in our field.
>
> Best,
>
> The authors

---

### Meta-Review · Area_Chair_SAqa · 2023-12-06

**Metareview:**

I sincerely thank the reviewing team and the authors for their engagement on this paper, which was well above average.

One of the major sticking points that the authors and the reviewers debated relates to this paper's "theoretical framework" positioning, which was challenged in different ways by all reviewers. Two of the most critical reviewers were UX5a and eRVD, with whom the authors had, at times, tense exchanges. Both reviewers had issues with the conceptual construction of the paper.

The authors tried to defend themselves by explaining that most of the misunderstandings came from a lack of knowledge of eRVD regarding the specifics of the area, as well as a different accepted standard of rigor (I note here that the reviewer is an expert in an adjacent area). I think both defenses are deficient, both in the principles and in the merits in this case. In fact, all reviewers had reservations regarding the exposition and the presentation of the paper. Second, the authors conceded on several important points brought up by reviewer eRVD.

I find the questions surrounding Point 3 of eRVD from the second half of the discussion (starting Nov 17th) especially important. Put concisely: it seems that the theoretical arguments on the paper fundamentally rely on assuming that a jointly optimal equilibrium is reached (an argument of the form "if good thing happens, more good things follow"); yet, the realization that the good premise cannot be achieved is somehow the starting point of the whole idea of ad-hoc teamwork, thus eroding the very foundations of the argument.

The author's response on this point is as follows: "The objective is "if we achieve the thing that we just said is not a reasonable expectation (joint optimality), then we achieve a good outcome (strict core stability)". Then, what we do is to enable the learner to influence other agents to achieve the optimal joint policy as much as possible, so as to achieve a good outcome.". While I can understand this objective, I agree with eRVD that the way the paper is presented is not very upfront about the disconnect between the starting point of the theory ("if we have joint optimality"), and the practical motivation of the area ("achieving joint optimality is not something we can hope for in general").

Overall, I would like to stress out that the issues the reviewers found on the paper pertain to the claims regarding the theoretical framework (which the reviewers find are not supported by what the results actually show), as well as the presentation. I believe that once those issues are improved, the paper will be very positively received. I encourage the authors to use the extensive discussion to guide a revision that can be enjoyed not just by experts on the topic, but also by experts in closely adjacent areas, taking the time to build bridges and cater to those as well, rather than building wall.

**Justification For Why Not Higher Score:**

The paper requires a substantial revision regarding presentation and theoretical claims

**Justification For Why Not Lower Score:**

N/A

---

### Decision · Program_Chairs · 2024-01-16

Reject